# *d1*: Scaling Reasoning in Diffusion Large Language Models via Reinforcement Learning

**Siyan Zhao**[*]
UCLA

**Devaansh Gupta**[*]
UCLA

**Qinqing Zheng**[†]
Meta AI

**Aditya Grover**[†]
UCLA

## Abstract

Recent large language models (LLMs) have demonstrated strong reasoning capabilities that benefits from online reinforcement learning (RL). These capabilities have primarily been demonstrated within the left-to-right autoregressive (AR) generation paradigm. In contrast, non-autoregressive paradigms based on diffusion generate text in a coarse-to-fine manner. Although recent diffusion-based large language models (dLLMs) have achieved competitive language modeling performance compared to their AR counterparts, it remains unclear if dLLMs can also leverage recent advances in LLM reasoning. To this end, we propose *d1*, a framework to adapt pre-trained masked dLLMs into reasoning models via a combination of supervised finetuning (SFT) and RL. Specifically, we develop and extend techniques to improve reasoning in pretrained dLLMs: (a) we utilize a masked SFT technique to distill knowledge and instill self-improvement behavior directly from existing datasets, and (b) we introduce a novel critic-free, policy-gradient based RL algorithm called *diffu*-GRPO, the first integration of policy gradient methods to masked dLLMs. Through empirical studies, we investigate the performance of different post-training recipes on multiple mathematical and planning benchmarks. We find that *d1* yields the best performance and significantly improves performance of a state-of-the-art dLLM. Our code is released at https://dllm-reasoning.github.io/.

## 1 Introduction

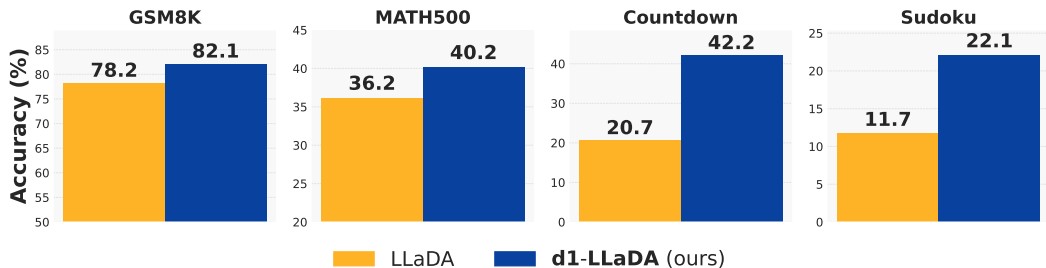

Figure 1: Across four math and planning tasks, **d1-LLaDA**, which undergoes SFT followed by our proposed ***diffu*-GRPO**, consistently outperforms the base LLaDA-8B-Instruct model. We report results using the best performing generation sequence length for each task and model, with complete sequence length results shown in Table 1.

---

[*]Equal contribution.

[†]Equal advising.

39th Conference on Neural Information Processing Systems (NeurIPS 2025).

Recent advances in large language models (LLMs) have demonstrated remarkable capabilities across diverse applications spanning chatbots, coding, summarization, and translation [1, 13]. While these models typically scale through next-token prediction on vast corpora via computationally intensive pretraining, the finite availability of high-quality training data poses a fundamental scaling challenge. Reinforcement learning (RL) methods have emerged as a promising post-training method, enabling models to generate and explore with reward signals rather than relying solely on static datasets. This approach has yielded significant improvements on reasoning tasks in recent models, such as DeepSeek-R1 [17] and Kimi K1.5 [41], demonstrating that applying RL directly to base models can achieve performance comparable to OpenAI's o1 model [31]. However, these advances in RL-based post-training have primarily been limited to autoregressive LLMs that operate through left-to-right, sequential inference.

In a parallel line of work, discrete diffusion large language models (dLLMs) [30, 15, 29, 48] have emerged as promising non-autoregressive alternatives for language modeling. Unlike AR models that generate text token-by-token in a causal manner, dLLMs generate text through an iterative denoising process, refining sequences over multiple steps while leveraging both past and future context via bidirectional attention. Among them, open masked dLLMs such as LLaDA [30] have demonstrated performance comparable to similarly sized AR models, and closed-source dLLMs such as Mercury [20] further demonstrate excellent inference efficiency. However, leading open-source dLLMs have not undergone RL post-training, leaving this promising direction largely unexplored. This paradigm shift raises important questions about how RL post-training might be effectively realized in a non-autoregressive context.

Adapting RL algorithms to masked dLLMs poses unique challenges because existing successful approaches for AR models, such as PPO [37] and GRPO [38], rely on estimating and optimizing policy distributions through computing log-probabilities of generated sequences, which cannot be directly applied to dLLMs. While this computation is straightforward in AR models through sequential factorization, dLLMs lack this natural decomposition due to their iterative, non-sequential generation process.

To bridge this gap, **we propose d1, a two-stage post-training framework for enhancing reasoning in masked dLLMs**. In the first stage, the model undergoes supervised finetuning (SFT) on high-quality reasoning traces. **In the RL stage, we introduce *diffu*-GRPO**, a novel policy gradient method for masked dLLMs that builds upon GRPO with our proposed efficient one-step estimation of log-probabilities. To the best of our knowledge, this represents the first application of policy gradient RL to masked dLLMs. Our estimator leverages random prompt masking, which acts a form of regularization for policy optimization, allowing us to scale the number of gradient updates per batch and reduces the number of online generations required by RL training. This substantially reduces the compute time.

Empirically, we instantiate d1 using LLaDA-8B-Instruct as our base model. We compare d1-LLaDA's performance with the base LLaDA model, as well as with LLaDA variants trained using SFT-only and *diffu*-GRPO-only approaches. Our experiments demonstrate that d1 consistently outperforms the base model across four reasoning tasks in math and planning, as shown in Figure 1, with nearly doubled performance on planning tasks. Furthermore, d1 surpasses both the SFT-only and *diffu*-GRPO-only methods. Additionally, we complement our primary findings with thorough ablation studies on algorithm design, qualitative analysis, and extensions of *diffu*-GRPO to coding tasks, where we also observe consistent improvements.

## 2 Preliminaries

### 2.1 Masked Diffusion Large Language Models

Masked dLLMs [5, 36, 39, 32, 26], involve a forward process that gradually corrupts a sequence of tokens $x_0$ by the `mask` token. The process is indexed by time $t \in [0, 1]$. At timestep $t$, the sequence $x_t$ is partially masked, where for each token the probability of remaining unmasked is $\alpha_t$. Particularly, $\alpha_t$ (a.k.a noise schedule) is strictly decreasing in $t$. When $t = 1$, all the tokens in $x_1$ are masked. To train a masked dLLM, we begin by designing a forward process with a specific form of $\alpha_t$. We parameterize a bidirectional unmasking predictor $f_\theta$. In each iteration, we randomly sample a timestep $t \in [0, 1)$ and mask the tokens based on the designed forward process. Given these

corrupted inputs, the learning objective is to predict the original tokens. The standard loss function for this task is the negative evidence lower bound (NELBO), which is an upper bound of the negative log-likelihood (NLL) of the data. For masked dLLMs, NELBO simplifies to a weighted NLL, where the weights are determined by a transformation of $\alpha_t$ [36, Equation (10)]. In this work, we apply d1 on top of LLaDA [30], whose forward process sets $\alpha_t = 1 - t$ and the resulting NELBO is

$$-\mathbb{E}_{t\sim\mathcal{U}[0,1),\ x_0\sim p_{\text{data}},\ x_t\sim q_{t|0}(x_t|x_0)} \left[ \frac{1}{t} \sum_{k=1}^{|x_t|} \mathbb{1}[x_t^k = \texttt{mask}] \log f_\theta(x_0^k \mid x_t) \right], \tag{1}$$

where $|x_t|$ is the sequence length of $x$, and $x^k$ is the $k$-th token. Note that the loss is only calculated for tokens that are masked out in timestep $t$. The key difference between masked dLLMs and BERT [12] is that the latter uses a fixed masking ratio and the decoding is a single-step infilling process, whereas masked dLLMs use time-varying masking ratios and the decoding process involves multiple steps starting from pure noise and thus resulting in a generative model. Further details about the formulation of masked dLLMs are deferred to Appendix C.

## 2.2 Group Relative Policy Optimization for Large Language Models

Policy gradient methods have been widely adopted in the post-training stage to enhance the performance of LLMs [33, 7, 22, 2]. While Proximal Policy Optimization (PPO) [37] has been the predominant approach in online RL, it requires jointly training a state value function $V$ to estimate advantages, leading to increased computational demands. Group Relative Policy Optimization (GRPO) [38] offers a more efficient alternative by using group statistics to derive advantages. For each question $q$, GRPO samples a group of $G$ responses $\{o_1, o_2, \ldots, o_G\}$ from the old policy $\pi_{\theta_{\text{old}}}$. It then sets the advantages for all tokens $k = 1, \ldots, |o_i|$ for $o_i$ as the normalized reward $\frac{r_i - \text{mean}(\{r_j\}_{j=1}^G)}{\text{std}(\{r_j\}_{j=1}^G)}$. Here, we can view $\text{mean}(\{r_j\}_{j=1}^G)$ as a $G$-sample Monte Carlo estimation of the value $V(q)$, while the sparse reward $r_i$ serves as the (undiscounted) state-action value $Q(q, o_i)$. However, normalizing the advantage $Q(q, o_i) - V(q)$ by nonzero state function introduces bias into policy gradient estimation. Therefore, similar to Liu et al. [24], we use the unnormalized advantage

$$A_i^k(\pi) = r_i(\pi) - \text{mean}(\{r_j(\pi)\}_{j=1}^G), \ 1 \leq k \leq |o_i|. \tag{2}$$

The rest of our RL setup follows GRPO. The objective function incorporates a clipping mechanism (similar to PPO) to moderate policy updates, and a reverse KL penalty to prevent excessive deviation from the reference policy:

$$\mathcal{L}_{\text{GRPO}}(\theta) = \mathbb{E}_{\substack{q\sim\mathcal{D} \\ o_1,\ldots,o_G\sim\pi_\theta(\cdot|q)}} \left[ \left( \frac{1}{G}\sum_{i=1}^G \frac{1}{|o_i|}\sum_{k=1}^{|o_i|} \min\left(\rho_i^k A_i^k, \text{clip}\left(\rho_i^k, 1-\varepsilon, 1+\varepsilon\right) A_i^k\right) \right) - \beta D_{\text{KL}}\left[\pi_\theta(\cdot|q)\|\pi_{\text{ref}}(\cdot|q)\right] \right], \tag{3}$$

where $\pi_\theta$ is the current policy being updated, $\pi_{\theta_{\text{old}}}$ is the policy before the update, $\rho_i^k = \frac{\pi_\theta(o_i^k|q, o_i^{<k})}{\pi_{\theta_{\text{old}}}(o_i^k|q, o_i^{<k})}$, $A_i^k$ is computed using $\pi_{\theta_{\text{old}}}$ and Equation (2), and $\pi_{\text{ref}}$ is the reference policy (typically the initial model). The clipping parameter $\varepsilon$ limits the magnitude of policy updates to ensure stability, while $\beta$ controls the strength of the KL divergence regularization.

## 3 d1: Adapting Pre-trained Masked dLLMs to Reasoning Models

We propose d1, a two-stage framework that enhances the reasoning performance of pre-trained masked dLLMs by sequentially combining SFT and online RL.

Online RL, particularly the GRPO algorithm, has demonstrated its efficacy in improving the performance of offline trained language model [38, 17, 41]. However, the learning formulation of GRPO does not directly generalize to dLLMs. The objective of GRPO (3) requires computing the (log-)likelihood ratio of $\pi_\theta$ and $\pi_{\theta_{\text{old}}}$, at both the token level (for the advantage weights) and the sequence level (for the reverse KL term). Generally speaking, we need to efficiently compute the per-token and the sequence log-probability of dLLMs' completion $o$. Autoregressive (AR) models, such as Transformers, directly model the per-token log-probabilities, and the sequence-level log-probability of $o$ can be easily computed through the chain rule using one forward pass: $\log \pi_{\text{AR}}(o|q) = \sum_{k=1}^{|o|} \log \pi_{\text{AR}}(o^k|q, o^{<k})$. Similarly, the KL term can be decomposed

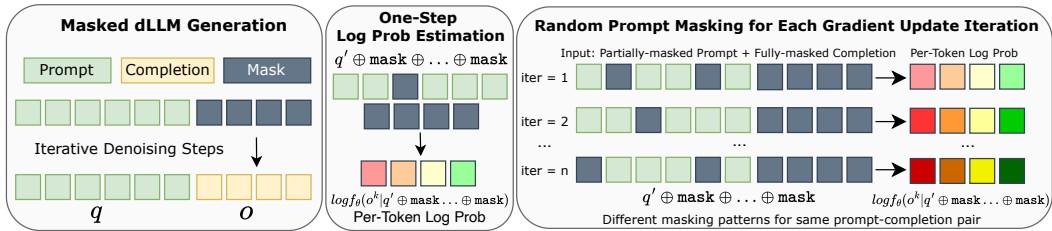

Figure 2: **Log Probability Estimation in *diffu*-GRPO.** After generating completion $o$ from prompt $q$ using full diffusion denoising (left), we compute token-level log probabilities with a single forward pass per masking pattern (mid) and use the log-probability of one-step unmasking as our estimation. During each policy gradient update, we apply a random masking pattern to the prompt, creating $q'$, while keeping the completion fully masked (right). The gradient of colors in the per-token log probabilities demonstrates that each distinct masking pattern yields a different estimate of the per-token log probabilities. This serves as a form of regularization for policy optimization, allowing more gradient updates per batch and thereby reducing the number of online generations needed for RL training.

as $D_{\text{KL}}\left[\pi_\theta(\cdot|q)\|\pi_{\text{ref}}(\cdot|q)\right] = \mathbb{E}_{o\sim\pi_\theta(\cdot|q)}\left[\sum_{k=1}^{|o_i|}\log\frac{\pi_\theta(o^k|q,o^{<k})}{\pi_{\text{ref}}(o^k|q,o^{<k})}\right]$. Unlike AR models, dLLMs do not adhere to sequential factorization of the sequence log-probability. Meanwhile, the per-token log-probability are also costly to compute since the decoding process invokes the unmasking predictor $f_\theta$ multiple times[3]. As the first step, we propose an efficient log-probability estimator in Section 3.1. Next, using these estimators, we introduce *diffu*-GRPO, a variant of GRPO for dLLMs in Section 3.2. Last, we discuss our SFT recipe in Section 3.3.

## 3.1 Efficient Log Probability Estimation for Masked dLLMs

For sequence log-probability, we use a mean-field approximation that decomposes it into a product of independent per-token log-probabilities. For per-token log-probability, we introduce an estimation method that only calls $f_\theta$ once.

**Mean-Field Approximation of Sequence Log Probability.** As opposed to AR models, dLLMs treat the token sequence as a whole and therefore its sequence-level log-probability lacks the AR decomposition. To efficiently estimate it, we use a simple mean-field decomposition to approximate $\log\pi_\theta(o|q)$ by $\sum_{k=1}^{|o|}\log\pi_\theta(o^k|q)$. The per-token log-probability estimation is introduced below.

**One-Step Per-Token Log Probability Estimation with Prompt Masking.** Let $\oplus$ denote the concatenation operator. Given a prompt $q$, the decoding process starts from an initial sequence $q\oplus\text{mask}\oplus\ldots\oplus\text{mask}$ (up to a preset length). To compute the log-probability of $o$, we perturb $q$ where every token is randomly masked out with probability $p_{\text{mask}}$, resulting in a new prompt $q'$. We then do one-step unmasking to obtain $\log f_\theta(o^k|q'\oplus\text{mask}\ldots\oplus\text{mask})$ and use it as an estimation of $\log\pi_\theta(o^k|q)$, $1\le k\le|o|$. We discuss the motivation of using a masked prompt $q'$ in the next section.

We note that LLaDA [30, Algorithm 3] uses a Monte Carlo type of approximation to estimate the log-probabilities, where they use a MC sample size is 128. This estimator is inefficient for online RL, since it creates a large computational graph with hundreds of forward passes, resulting in inefficient policy optimization and excessive memory usage.

## 3.2 *diffu*-GRPO: Policy Gradient Optimization for Masked dLLMs

Using the log-probability estimators proposed in Section 3.1, we extend GRPO to masked dLLMs. Note that our estimation technique is broadly applicable and can readily extend to other policy gradient methods such as PPO [37] or REINFORCE [44].

---

[3]In other words, $\pi_\theta$ is a composition of $M$ $f_\theta$ functions for a $M$-step decoding process

---

**Algorithm 1** *diffu*-GRPO: Policy Gradient Optimization for Masked dLLMs

---

**Require:** Reference model $\pi_{\text{ref}}$, prompt distribution $\mathcal{D}$, number of completions per prompt $G$, number of inner updates $\mu$, prompt token masking probability $p_{\text{mask}}$
 1: Initialize $\pi_\theta \leftarrow \pi_{\text{ref}}$
 2: **while** not converged **do**
 3:     $\pi_{\theta_{\text{old}}} \leftarrow \pi_\theta$
 4:     Sample a prompt $q \sim \mathcal{D}$
 5:     Sample $G$ completions $o_i \sim \pi_{\theta_{\text{old}}}(\cdot \mid q), i \in [G]$
 6:     For each $o_i$, compute reward $r_i$ and advantage $A_i^k(\pi_{\theta_{\text{old}}})$ using Equation (2)
 7:     **for** gradient update iterations $n = 1, \ldots, \mu$ **do**
 8:         $q' \leftarrow$ randomly mask tokens of prompt $p$ with probability $p_{\text{mask}}$
 9:         For $\pi_\theta, \pi_{\theta_{\text{old}}}, \pi_{\text{ref}}$, estimate log-probabilities of $o_i$ given $q'$ according to Section 3.1
10:         Compute *diffu*-GRPO objective (4) and update $\pi_\theta$ by gradient descent
11: **return** $\pi_\theta$

---

Let $\phi^{\pi_\theta}(o^k \mid q')$ and $\phi^{\pi_\theta}(o \mid q')$ denote the estimated per-token and sequence probabilities for $\pi_\theta$. We derive the loss function of *diffu*-GRPO,

$$
\mathcal{L}_{\textit{diffu}\text{-GRPO}}(\theta) = \mathbb{E}_{\substack{q\sim\mathcal{D},\, q'\sim\text{masking}(q),\\ o_1,\ldots,o_G\sim\pi_{\theta_{\text{old}}}(\cdot|q)}} \left[ \frac{1}{G}\sum_{i=1}^{G}\frac{1}{|o_i|}\sum_{k=1}^{|o_i|}\min\left( \frac{\phi^{\pi_\theta}(o_i^k \mid q')}{\phi^{\pi_{\theta_{\text{old}}}}(o_i^k \mid q')}A_i^k, \right.\right.
$$
$$
\left.\left. \text{clip}\left( \frac{\phi^{\pi_\theta}(o_i^k \mid q')}{\phi^{\pi_{\theta_{\text{old}}}}(o_i^k \mid q')}, 1-\varepsilon,\, 1+\varepsilon \right)A_i^k \right) - \beta\, D_{\text{KL}}\Big[ \phi^{\pi_\theta}(\cdot \mid q') \,\big\|\, \phi^{\pi_{\text{ref}}}(\cdot \mid q') \Big] \right] \tag{4}
$$

Our algorithm is summarized in Algorithm 1. To efficiently optimize the policy loss, in practice, on-policy RL algorithms such as PPO and GRPO perform multiple gradient updates for each batch of samples. During these updates, the prompt $q$, completions $\{o_i\}_{i=1}^{G}$, old policy $\pi_{\theta_{\text{old}}}$ and advantages $A_i^k(\pi_{\theta_{\text{old}}})$ are kept fixed. However, determining the optimal number of gradient updates per batch is challenging. If the number is too high, it can lead to overfitting within the batch, while a number that is too low slows down convergence. Achieving a balance between outer batch iterations and inner gradient updates is crucial for sample efficiency. Besides, every outer batch iteration requires sampling completion through iterative denoising steps, which incurs high computational cost.

Interestingly, our log-probability estimator offers a unique mitigation to this dilemma. For each gradient update step, we randomly mask the prompt $q$ to $q'$ to estimate the log-probabilities. Intuitively, this stochastic masking introduces perturbed views of the same (prompt, completion) pairs, serving as a form of regularization for policy optimization. It can also be viewed as a form of data augmentation, extracting more supervision signals from the same data. Empirically, we found that this approach, unique to masked diffusion models, allows us to scale $\mu$ to higher values while maintaining stable learning dynamics. As a consequence, it reduces the number of outer batch iterations required for convergence, which in turn decreases the number of online generations needed and ultimately results in significantly lower computational cost. As shown in Figure 5, training with higher values of $\mu$ achieves the same reward performance in substantially less wall clock time.

### 3.3 Supervised FineTuning with Reasoning Data

We perform SFT of LLaDA on s1K [28], a curated dataset consisting of 1000 high-quality reasoning questions. The reasoning traces in s1K exhibit detailed step-by-step problem-solving processes, including verification of intermediate results and backtracking when encountering errors or dead ends. The SFT algorithm is summarized in Algorithm 2, where tokens are randomly masked during training according to a time-varying schedule. The model is optimized to predict the original tokens given their context. We find that for SFT to work effectively in practice, various design choices must be carefully considered, whose details are discussed in Appendix D.2.

## 4 Experiments

To understand how reasoning capabilities can be scaled in masked dLLMs through training adaptations, we conduct comprehensive experiments to answer the following main research questions:

Table 1: **Model performance on Mathematics and Planning Benchmarks:** **Green values** indicate best performance and **blue values** indicate second-best performance. The results demonstrate that **d1-LLaDA** consistently outperforms other models, applying *diffu*-GRPO consistently improves the starting checkpoint, and *diffu*-GRPO alone shows better performance than SFT.

| Model / Seq Len | GSM8K | | | MATH500 | | | Countdown | | | Sudoku | | |
|---|---|---|---|---|---|---|---|---|---|---|---|---|
| | 128 | 256 | 512 | 128 | 256 | 512 | 128 | 256 | 512 | 128 | 256 | 512 |
| LLaDA-8B-Instruct | 68.7 | 76.7 | 78.2 | 26.0 | 32.4 | 36.2 | 20.7 | 19.5 | 16.0 | 11.7 | 6.7 | 5.5 |
| + SFT | 66.5 | 78.8 | 81.1 | 26.2 | 32.6 | 34.8 | 20.3 | 14.5 | 23.8 | 16.5 | 8.5 | 4.6 |
| + *diffu*-GRPO | 72.6 | 79.8 | 81.9 | 33.2 | 37.2 | 39.2 | 33.2 | 31.3 | 37.1 | 18.4 | 12.9 | **11.0** |
| + SFT + *diffu*-GRPO (**d1-LLaDA**) | **73.2** | **81.1** | **82.1** | **33.8** | **38.6** | **40.2** | **34.8** | **32.0** | **42.2** | **22.1** | **16.7** | 9.5 |

(1) How do SFT on reasoning traces and applying *diffu*-GRPO ***independently*** improve LLaDA's reasoning capabilities?

(2) What additional gains can be achieved by *combining* SFT and *diffu*-GRPO to create d1-LLaDA?

(3) **Design Choices:** How does the proposed log-probability estimation with *randomized masking* in *diffu*-GRPO and the masking probability $p_{mask}$ affect training efficiency and stability?

## 4.1 Models, Tasks and Setups

**Models** We employ LLaDA-8B-Instruct [30], a state-of-the-art open-sourced dLLM that has not undergone post-training, as our primary experimental testbed and baseline. We apply 3 post-training recipes to LLaDA-8B-Instruct: (a) SFT, (b) *diffu*-GRPO, (c) d1: applying *diffu*-GRPO on the checkpoint after SFT, where we refer to them as LLaDA+SFT, LLaDA+*diffu*-GRPO, and d1-LLaDA, respectively.

**Tasks** We conduct experiments on six reasoning tasks in three categories: (1) **Mathematical reasoning**: we use GSM8K [10], a dataset of multi-step grade school math problems, and MATH500 [23], a curated subset of 500 problems drawn from the MATH dataset [18] comprising high-school competition math problems; (2) **Planning**: this includes two tasks: 4x4 Sudoku puzzles, which require constraint satisfaction and systematic elimination to fill a grid with numbers; and Countdown with 3 numbers, a combinatorial arithmetic game in which models must reach target numbers using basic arithmetic operations on a given set of numbers. (3) **Coding**: comprises of two benchmarks; HumanEval [8], a suite of 164 hand-crafted Python algorithmic programming problems and MBPP [6], a crowd-sourced collection of 257 Python tasks.

**Training** For SFT, we train on s1k [28] for 20 epochs, with a sequence length of 4096. For RL, we train a separate model for each task. More specifically, for GSM8K, MATH500, we train on the training split; for Countdown and Sudoku, we train on synthetic generated datasets. We use a composed reward function that combines both formatting and correctness rewards. Due to the heavy computational cost of online generations, we limit the generation sequence length of online generations to be 256 throughout RL training. Other hyperparameters of training, training and evaluation datasets, reward functions, and inference setups are detailed in Appendix D.

**Evaluation** For all the benchmarks, we evaluate LLaDA-8B-Instruct and LLaDA+SFT on the final checkpoint for all the tasks. For LLaDA+*diffu*-GRPO and d1-LLaDA, we evaluate every 100 steps starting from step 600 and report the best results. We evaluate all models with 0-shot-prompting and greedy decoding with generation lengths of 128, 256 and 512 separately.

## 4.2 Main Results

***diffu*-GRPO outperforms both LLaDA and SFT and improves over initialization checkpoint consistently .** Table 1 reports the performance of baseline LLaDA-8B-Instruct and models obtained by different post-training recipes across four tasks using zero-shot evaluation, where each *diffu*-GRPO model was trained for each task. For each task, we evaluate with three generation sequence lengths, and Figure 4 plots the average number of effective tokens. We present the following predominent findings.

Both *diffu*-GRPO and SFT yield improvements over the LLaDA-8B-Instruct baseline, with *diffu*-GRPO demonstrating consistently larger gains. Specifically, *diffu*-GRPO outperforms both LLaDA-8B-Instruct and SFT, in all 12 setups, while SFT outperforms LLaDA-8B-Instruct in only 7 of

them, demonstrating that *diffu*-GRPO achieves stronger overall performance than SFT alone. Both LLaDA+*diffu*-GRPO and d1-LLaDA demonstrate consistent improvements over their respective starting points. Specifically, LLaDA+*diffu*-GRPO outperforms the base LLaDA-8B-Instruct model across all setups, and d1-LLaDA surpasses LLaDA+SFT in every case. This indicates that *diffu*-GRPO provides reliable performance gains, regardless of the initialization—whether from a pretrained model or an SFT-adapted checkpoint.

**d1 recipe yields the highest gains.** SFT, followed by *diffu*-GRPO—resulting in d1-LLaDA—yields additional gains, beyond either method individually. This combined approach outperforms pure *diffu*-GRPO in 11 out of 12 setups, indicating a synergistic effect between the two training stages. Notably, while d1-LLaDA shows consistent improvements across all benchmarks, the magnitude varies by task: we observe modest improvements on GSM8K (3.9%) and MATH500 (4.0%), but significantly larger gains on Countdown (26.2%) and Sudoku (10.0%). We hypothesize this discrepancy stems from the base model's saturation on mathematical tasks, with less room for improvement as compared to planning benchmarks that involve structured constraint satisfaction patterns.

**Training a unified model across tasks retains strong performance.** We train a single *diffu*-GRPO (and d1) model on the *combined* GSM8K, MATH500, Countdown, and Sudoku datasets. To ensure balanced training, we subsample the data so that each task has the same number of training examples. Even with subsampling, Table 2 shows that *diffu*-GRPO scales well to multi-task settings without sacrificing accuracy compared to the per-task *diffu*-GRPO results in Table 1.

**Scaling *diffu*-GRPO to coding domains.** We also evaluate *diffu*-GRPO on coding tasks, where we train a model on the KodCode-Light-RL-10K dataset [45], which contains general coding tasks with solutions verified by synthetic unit tests. The *diffu*-GRPO results are shown in Table 3. We find that *diffu*-GRPO consistently improves performance, regardless of the initialization point. Interestingly, our findings suggest that s1k is not suitable for coding, since it **lacks datapoints with code**. Exploration into finding the optimal SFT dataset is left for future works.

Table 2: **Unified Model Performance Across Reasoning Tasks:** For *diffu*-GRPO and d1-LLaDA variants, a single model was trained on the combined dataset of GSM8K, MATH500, Countdown, and Sudoku. **Green** and **blue values** indicate the best and second-best performance.

| | GSM8K | | MATH500 | | Countdown | | Sudoku | |
|---|---|---|---|---|---|---|---|---|
| Model / Seq Len | 128 | 256 | 128 | 256 | 128 | 256 | 128 | 256 |
| LLaDA-8B-Instruct | 68.7 | 76.7 | 26.0 | 32.4 | 20.7 | 19.5 | 11.7 | 6.7 |
| + SFT (s1k) | 66.5 | 78.8 | 26.2 | 32.6 | 20.3 | 14.5 | 16.5 | 8.5 |
| + combined *diffu*-GRPO | **72.4** | 78.2 | **30.2** | **36.6** | 27.7 | 19.5 | **22.9** | **15.7** |
| combined d1-LLaDA | **75.1** | **81.1** | 29.8 | 35.4 | **30.1** | **32.8** | 21.9 | 15.4 |

Table 3: **Effectiveness of *diffu*-GRPO on Coding Benchmarks:** Evaluated with and without *diffu*-GRPO on HumanEval and MBPP. *diffu*-GRPO consistently improves over initialization checkpoint on coding tasks.

| | HumanEval | | | MBPP | | |
|---|---|---|---|---|---|---|
| Model / Seq Len | 128 | 256 | 512 | 128 | 256 | 512 |
| **LLaDA-8B-Instruct** | 27.4 | 35.3 | 37.8 | 36.2 | 41.2 | 40.4 |
| + *diffu*-GRPO | 29.3 | 39.0 | 34.8 | 42.0 | 45.5 | 41.6 |
| Δ (*diffu*-GRPO gain) | **+1.9** | **+3.7** | **-3.0** | **+5.8** | **+4.3** | **+1.2** |
| **LLaDA-8B-Instruct + SFT (s1k)** | 21.3 | 32.3 | 32.9 | 40.1 | 39.7 | 41.2 |
| + *diffu*-GRPO | 31.1 | 32.9 | 37.8 | 40.5 | 44.7 | 42.8 |
| Δ (*diffu*-GRPO gain) | **+9.8** | **+0.6** | **+4.9** | **+0.4** | **+5.0** | **+1.6** |

***diffu*-GRPO improves reasoning beyond training sequence length.** Although our *diffu*-GRPO training uses fixed sequence length of 256 for online generations, we observe performance gains at other generation sequence lengths as well. The improvements at 128 and 512 sequence lengths suggest that the model has learned more general reasoning strategies rather than overfitting to a specific length. This is further supported by the effective token usage data, presented in Figure 4, which shows no truncation at 128 tokens and increased token utilization at 512.

## 4.3 Discussion

**Qualitative results show "aha moments" in SFT and d1-LLaDA generations.** While the performance for generation sequence length 128 and 256 increases with SFT, *diffu*-GRPO and d1 as compared to LLaDA-8B-Instruct, qualitatively, we do not observe significant differences in the generated reasoning traces. However, at sequence length 512, we begin observing "aha moments" in the SFT and d1-LLaDA models, which demonstrates self-correction and backtracking behaviors. We show these in Appendix E. For the same questions from GSM8k, we show generations of each model, with the variants using SFT showing self-verifications and self-corrections to the right answer. Our intuition is that the model has instilled behaviors such as verification of intermediate results and backtracking from the reasoning traces of s1k during the SFT stage.

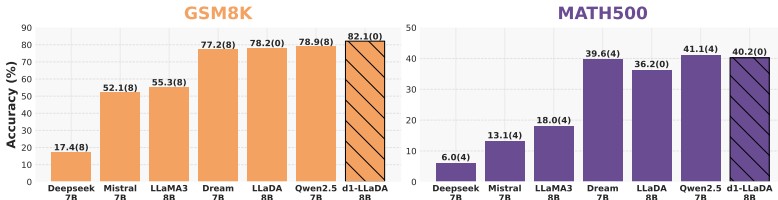

Figure 3: **Comparison with state-of-the-art dLLMs and AR LLMs of similar size:** d1-LLaDA achieves the highest GSM8K score and the second-highest MATH500 score. LLaDA results are from our evaluation using 0-shot. Scores for other models are from Dream [48], using 8-shot prompts for GSM8K and 4-shot for MATH. Note that here we report d1-LLaDA with task-specific RL training.

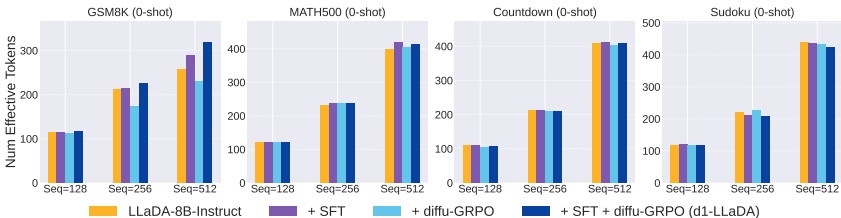

Figure 4: **Effective Token Usage:** As we increase the evaluation generation length, the number of effective tokens (average number of non-padding, non-EOS tokens per generation across tasks) grows and remains comparable for all the methods on MATH500, Countdown and Sudoku tasks.

**Sequential scaling with increasing generation sequence lengths.** LLaDA-8B-Instruct, SFT, *diffu*-GRPO and d1-LLaDA demonstrate improved performance with increasing sequence lengths for GSM8k and MATH500, with larger jumps observed from 128 to 256 ($\sim 7.1\%$), than from 256 to 512 ($\sim 2.5\%$). Qualitative examples in Appendix E show more sophisticated reasoning traces emerge with 512-token generation lengths. These findings align with previous research showing that increasing test-time compute through longer reasoning processes leads to improved performance in autoregressive models [28]. However, we notice a mixed scaling trend on Countdown and Sudoku. Performance decreases with increasing sequence lengths for Sudoku across all models. For Countdown, LLaDA-8B-Instruct decreases monotonically with sequence length, while SFT, *diffu*-GRPO and d1-LLaDA peak at 512 sequence length. This likely stems from extensive searching requirements, beyond LLaDA-8B-Instruct's capabilities. We hypothesize favorable sequential scaling will strengthen with more robust base dLLMs. Unlike AR models like DeepSeek R1 [17], we observe no significant CoT length growth post-RL training, as LLaDA-8B-Instruct was pre-trained on sequences up to 4096 tokens. Further scaling requires larger generation lengths during RL training, currently infeasible due to slow generation speed. Future research should develop efficient inference algorithms for online sampling to scale dLLM RL training.

### 4.4 Design Choices and Ablations for *diffu*-GRPO

**Random Masking for Likelihood Estimation Offers Implicit Regularization** Our randomized masking mechanism provides significant advantages for training masked dLLMs. As shown in Figure 5, random masking consistently outperforms fixed masking across different values of policy optimization updates ($\mu$). While conventional approaches typically limit $\mu$ to 2 due to diminishing returns and overfitting risks, our approach enables scaling $\mu$ to much higher values (12, or even 24) while maintaining or improving performance, facilitating faster convergence of RL training. Consequently, fewer number of generations are needed, which in turn remarkably reduces the computational cost. The rightmost plot demonstrates the real-world efficiency gains, where models with higher $\mu$ values achieve better correctness rewards in significantly lesser wall clock time. This efficiency stems from creating diverse views of the input data during each optimization step, allowing the model to prevent in-batch overfitting and extract more learning signal from each generation.

**Effect of Masking Rate on Training Stability and Performance** We examine how prompt masking probability $p_{mask}$ influences *diffu*-GRPO training. As shown in Figure 6, lower rates (0.1, 0.3) yield more stable training and better final performance by preserving more context tokens without masking

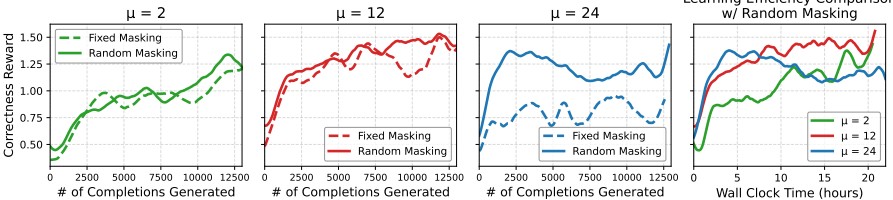

Figure 5: **Comparison of fixed vs. random masking** across different policy optimization update values ($\mu$). The first three figures show GSM8K correctness reward vs. the number of completions generated during RL training with different $\mu$. Random masking consistently outperforms fixed masking. The rightmost panel compares all three $\mu$ values with random masking in terms of wall clock time, indicating higher efficiency from higher $\mu$ values.

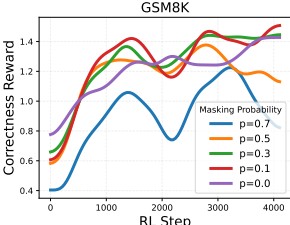

Figure 6: **Ablation of prompt masking probability ($p_{\mathbf{mask}}$)** on GSM8K reward trends. Light masking (0.1, 0.3) improves stability and performance over no masking (0.0), suggesting the regularization benefit of random masking as discussed in Sec 3.2. Higher masking rates (0.5, 0.7) introduce instability in later training stages.

them, while higher rates (0.5, 0.7) introduce instability, with 0.7 causing sharp degradation after 3000 steps. Although $p_{\text{mask}} = 0.0$ avoids variability, it underperforms slightly, confirming the regularization effect brought by random masking as discussed in Sec. 3.2. This effect is especially beneficial at large policy iteration counts ($\mu = 12$), as used in this ablation.

## 5 Related Works

Due to space constraint, we provide a detailed related works discussion in Appendix B.

**Diffusion Language Models.** Diffusion models, successful in visual domains [40, 19], faced challenges in language due to text's discrete nature, initially tackled by modeling continuous diffusion on textual latents [5, 16]. Masked diffusion emerged as an effective discrete variant [5, 36, 39, 32, 29], scaled notably in DiffuLLaMA [15], which initialized with pretrained LLaMA weights. Recent works explored chain-of-thought reasoning [47, 46], block-based generation [4], and large-scale competitive performance in LLaDA [30] and Dream [48]. However, reinforcement learning (RL) enhancement remains unexplored; we present the first demonstration using policy gradients for large diffusion language models. **Improving Reasoning Abilities of LLMs through SFT and RL.** Reasoning improvements in LLMs involve supervised finetuning (SFT) with high-quality reasoning datasets [50, 21, 35] or curated reasoning demonstrations [49, 28, 52]. However, RL approaches [9] generalize better, especially with methods like GRPO [17, 38], facilitating advantage estimation without critic models. Advanced reasoning via RL alone was shown by DeepSeek-R1-Zero [17], whose reasoning traces can be used to distill smaller-model, such as OpenThoughts [42] and OpenR1-Math[4]. Prior RL work in discrete diffusion models [51] employed concrete score matching and applied to smaller scale models, whereas our method specifically applies to large masked dLLMs with efficient masking-based policy gradients, integrating both SFT and RL.

## 6 Conclusion

In this work, we explore scaling reasoning in diffusion LLMs through different recipes. SFT on reasoning datasets improves performance and reveals "Aha moments". We introduce *diffu*-GRPO, an efficient policy gradient method for dLLMs that consistently outperforms SFT across benchmarks. Combining these approaches, our d1 recipe—a two-stage SFT and *diffu*-GRPO pipeline—delivers the most significant improvements over the baseline. Future work should focus on developing efficient decoding strategies to scale generation length for more effective RL training.

---

[4] https://huggingface.co/datasets/open-r1/OpenR1-Math-220k

## Acknowledgments

This research was supported by NSF CAREER Grant #2341040, a Schmidt AI 2050 Fellowship and a gift from Toyota.

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

# A Limitations

Due to the fixed-length generation requirement of LLaDA, our *diffu*-GRPO training is conducted with a predefined sequence length, which may constrain the model from discovering optimal reasoning paths—either concise solutions or extended chain-of-thought traces—as observed in prior autoregressive works like DeepSeek-R1. Future work could explore applying *diffu*-GRPO to models like Block Diffusion that support variable-length generation and enable scalable long-context RL training.

# B Related Work

**Diffusion Language Models** While diffusion models have achieved remarkable success in the visual domain [40, 19], their application to language has been limited, partly due to text's discrete nature. Initial approaches attempted to learn continuous diffusion models over textual latents [5, 16], but faced challenges with scalability and discretization. Masked diffusion has been established as a specific instance of discrete diffusion [5, 36, 39, 32, 29], with recent efforts scaling these models significantly. DiffuLLaMA [15] extended this approach by initializing masked diffusion language models with pretrained LLaMA weights. Ye et al. [47] explored how diffusion language models can generate chain-of-thought reasoning, and complex reasoning tasks on smaller-scale models [46], highlighting their advantages over autoregressive models in reversal tasks, though their traces lacked self-correction capabilities. Arriola et al. [4] proposed Block Diffusion, a hybrid approach that models sequences block-by-block while applying diffusion within each block, allowing flexible length generation and improving inference efficiency with kv-caching. Recently, LLaDA [30] and Dream [48] demonstrated that large diffusion language models can achieve performance comparable to similarly-sized autoregressive alternatives, but have not yet been enhanced through reinforcement learning. To the best of our knowledge, we are the first to demonstrate the efficacy of policy gradient-based reinforcement learning algorithms on large diffusion language models.

**Improving Reasoning Abilities of LLMs through SFT and RL** Approaches to enhance reasoning capabilities in large language models generally fall into two categories: supervised finetuning and reinforcement learning. SFT on high-quality reasoning traces [50, 21, 35] has shown promising results, while fewer but carefully curated reasoning datasets [49, 28, 52] can outperform larger datasets. Chu et al. [9] demonstrate that SFT-based reasoning often relies on memorization rather than generalization, while RL methods achieve better transfer to novel scenarios, particularly when intermediate reasoning steps are difficult to supervise. Recently, algorithms like GRPO [17, 38] enable efficient training by estimating advantages from group scores without requiring additional critic models as in PPO. Guo et al. [17] demonstrate that strong reasoning capabilities can emerge through RL even without SFT (DeepSeek-R1-Zero), producing long reasoning traces with self-reflection and verification steps that significantly improve performance on mathematical tasks. The development of strong reasoning models like R1 has in turn sparked renewed interest in SFT for smaller models using distilled reasoning traces from these expert reasoners. Datasets like OpenThoughts [42] and OpenR1-Math[5], which contain reasoning traces from DeepSeek R1, enable smaller models to learn step-by-step problem-solving from expert demonstrations. For RL in discrete diffusion models, prior work by Zekri and Boullé [51] proposed a policy gradient framework using concrete score matching, but it relies on gradient-flow computations and does not target masked objectives. In contrast, our method is tailored to masked dLLMs with efficient policy gradient calculation and improved learning efficiency through random masking. Our work is among the first to explore improving reasoning in diffusion-based LLMs via both SFT and RL.

---

[5] https://huggingface.co/datasets/open-r1/OpenR1-Math-220k

## C   Masked dLLM Formulation

Masked diffusion language model sequence of tokens $x_t, t \in [0, 1)$, which follow a forward diffusion process $q$. This process takes as input the complete sequence $x_0$ at $t = 0$ and gradually corrupts it by randomly replacing tokens with a mask token `mask`. Therefore, $x_t$ represents the sequence with increasing masking ratios in expectation. Each token in the sequence $x_t^i$ thus follows the conditional distribution,

$$q_{t|0}(x_t|x_0) = \prod_{i=0}^{L} q_{t|0}(x_t^i|x_0^i), \qquad q_{t|0}(x_t^i|x_0^i) = \begin{cases} 1 - \alpha_t, & x_t^i = \texttt{mask} \\ \alpha_t, & x_t^i = x_0^i \end{cases} \tag{5}$$

where $\alpha_t$ (a.k.a noise schedule) is strictly decreasing in $t$. Simply put, at any timestep, the probability that a token transitions to the masked state is $\alpha_t$. At the end of the forward process, i.e. at $t = 1$, all tokens are guaranteed to be masked.

This masked sequence serves as the input for the reverse process. A key property of the forward process is that once a token transitions to the masked state, it cannot transition to any other state. Therefore, the conditional distribution from an arbitrary time step $t$ to $s$ (i.e., the reverse process), such that $0 \leq s < t \leq 1$ is given by,

$$q_{s|t}(x_s^i|x_t) = \begin{cases} 1, & x_t^i \neq \texttt{mask}, \ x_s^i = x_t^i \\ \frac{1-\alpha_s}{1-\alpha_t}, & x_t^i = \texttt{mask}, \ x_s^i = \texttt{mask} \\ \frac{\alpha_s - \alpha_t}{1-\alpha_t} q_{0|t}(x_s^i|x_t), & x_t^i = \texttt{mask}, \ x_s^i \neq \texttt{mask} \\ 0, & \text{otherwise} \end{cases} \tag{6}$$

The function $q_{0|t}(x_s^i|x_t)$ is estimated by the language model, that predicts the original token in sequence $x_0$, if it is masked in $x_t$. Notably, previous works find that the model does not require the timestep as an input [] since the number of mask tokens implicitly provide this information to the model.

The model, parameterized as $f_\theta(\cdot|x_t)$ learns to predict all the masked tokens in the sequence $x_t$ simultaneously, similar to the masked language modeling task. More specifically, it is trained by minimizing a NELBO of the negative log-likelihood, given by,

$$\text{NELBO}(\theta) \triangleq \mathbb{E}_{x_0, x_t} \left[ \int_{t=0}^{t=1} \frac{\alpha_t'}{1-\alpha_t} \sum_{i=1}^{L} \mathbb{1}[x_t^i = \texttt{mask}] \log f_\theta(x_0^i \mid x_t) \right], \tag{7}$$

where $x_0$ is sampled from the training data distribution $p_{\text{data}}$, and $x_t \sim q_{t|0}(\cdot|x_0)$. In summary, the model is trained to reverse the forward process by gradually denoising (unmasking) the input sequence (all masked tokens) and recover the data distribution.

While various forms of noise schedules can be used [36, 39], Nie et al. [30, LLaDA] uses the linear schedule: $\alpha_t = 1 - t$. The resulting loss function is a specific form of Equation (7):

$$-\mathbb{E}_{t \sim \mathcal{U}[0,1], \, x_0, \, x_t} \left[ \frac{1}{t} \sum_{i=1}^{L} \mathbb{1}[x_t^i = \texttt{mask}] \log f_\theta(x_0^i \mid x_t) \right]. \tag{8}$$

# D Experiment Details

**Inference** To decode a sequence of $N$ tokens, we use $\frac{N}{2}$ denoising steps and unmask 2 tokens in each step. While the decoding process can generate tokens in any order, we find that decoding from left to right in blocks yields slightly better performance in practice. This is referred to as the semi-autoregressive decoding strategy [30]. More specifically, we divide the sequence into blocks of 32 tokens. In each step, we unmask 2 tokens with the highest confidence within the current block, regardless of their position. Once all the tokens in the current block are unmasked, we move to the next one.

## D.1 *diffu*-GRPO

We use the TRL library [43] to implement *diffu*-GRPO. For our *diffu*-GRPO training, we employed Low-Rank Adaptation (LoRA) with a rank of $r = 128$ and scaling factor $\alpha = 64$.

For *diffu*-GRPO on gsm8k, math, countdown and sudoku tasks, training was conducted on 8 NVIDIA A100-80G GPUs, with the following hyperparameters: sequence length of 256 tokens, batch size of 6 per GPU, and gradient accumulation steps of 2. We optimized the model using the AdamW optimizer [25], with parameters $\beta_1 = 0.9$, $\beta_2 = 0.99$, weight decay of 0.1, learning rate of $3 \times 10^{-6}$, and gradient clipping at 0.2. For computational efficiency, we utilized Flash Attention 2 [11] and 4-bit quantization. In gradient update iterations, each token in the prompt is randomly masked with a probability $p_{\text{mask}} = 0.15$ for log-probability estimation. Our codebase contains further configuration details: `https://github.com/dllm-reasoning/d1`. We train 7700, 6600 steps (number of gradient updates) for GSM8K and MATH500 respectively; for Countdown and Sudoku, we train on synthetic generated datasets for 5000, 3800 steps respectively.

For *diffu*-GRPO on coding task, training was conducted on 4 NVIDIA RTX A5000 for 7500 steps (base model + *diffu*-GRPO) and 9000 steps(SFT model + *diffu*-GRPO), with a per-device batch size of 2 and 4 gradient accumulation steps. The other hyperparameters remain the same as other tasks. Exact configuration details have been provided in our codebase.

### D.1.1 Reward Functions, RL Training, and Evaluation Datasets

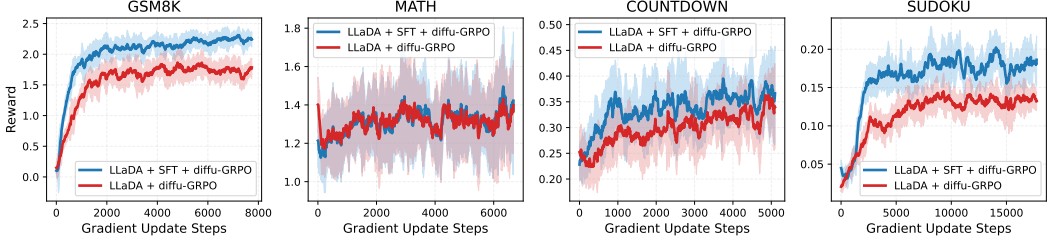

Figure 7: Reward curves during RL training for the models in Table 1, across four reasoning tasks. We compare LLaDA+*diffu*-GRPO and *d1*-LLaDA (+SFT + *diffu*-GRPO). *d1*-LLaDA consistently achieves higher or comparable reward trajectories.

We designed specific reward functions to guide the model's learning for each task. The rewards are structured to encourage proper formatting, accurate reasoning, and correct solutions, with varying levels of granularity depending on task requirements. We show the training curves of the results in Table 1 in Figure 7.

**GSM8K** For the GSM8K dataset, we conduct RL on the training split of the GSM8K dataset [6]and evaluate on the test split. We employ a composite reward function consisting of five components following the unsloth implementation of reward functions[7], we used these:

- **XML Structure Reward**: Rewards proper formatting with reasoning and answer tags:
  - +0.125 for each correctly placed opening and closing tag

---

[6]`https://huggingface.co/datasets/openai/gsm8k`
[7]`https://unsloth.ai/blog/r1-reasoning`

- – Small penalties for extraneous content after closing tags

- **Soft Format Reward**: Awards 0.5 points for responses matching the pattern:

  `<reasoning>...(content)...</reasoning><answer>...(content)...</answer>`

- **Strict Format Reward**: Awards 0.5 points for adhering to the exact prescribed format with appropriate line breaks.

- **Integer Answer Reward**: Awards 0.5 points if the extracted answer is a valid integer.

- **Correctness Reward**: Awards 2.0 points if the extracted answer exactly matches the ground truth.

**Countdown**  For the Countdown task, we train on the training split of the dataset[8] from the TinyZero project [34], restricting to instances that use only three numbers. And we evaluate on 256 synthetically generated countdown questions with 3 numbers. We implement a reward function that checks if an arithmetic expression constructed from given numbers reaches a target value:

The function awards:

- 1.0 point when the equation equals the target and uses exactly the available numbers
- 0.1 points when the equation uses the right numbers but doesn't reach the target
- 0 points otherwise

**Sudoku**  For the 4×4 Sudoku task, we utilize the training dataset available at `https://github.com/Black-Phoenix/4x4-Sudoku-Dataset`, specifically the subset containing one million unique puzzles. This dataset was synthetically generated using code from Arel [3]. For evaluation purposes, we randomly generate 256 Sudoku puzzles using this generator. The reward is calculated as the proportion of correctly filled cells among those that were empty in the original puzzle. This approach focuses evaluation on the model's problem-solving ability rather than its capacity to copy pre-filled values.

**MATH500**  For the MATH500 task, we train on the train split of the MATH dataset[9]. Like GSM8k, we employ a composite reward function consisting of:

- **Format Reward**: We award format reward points depending on the presence of <answer></answer> tags and \boxed, as follows:
  - 1.00 point if answer tags are present with \boxed inside them
  - 0.75 points if answer tags are present without \boxed in them
  - 0.50 points if answer tags are not present, but \boxed is present
  - 0.25 points if neither answer tags, nor \boxed is present
- **Correctness Reward**: 2.0 points if the correct answer is in \boxed{}

**Coding**  For the coding model, we train on the KodCode-Light-RL-10k[10] dataset. Again, we use a composite reward function comprising of:

- **XML Structure Reward:** The same function used for GSM8k is also used for this task, with the addition that an extra 0.5 points are provided if the program is within answer tags. Additionally, 0 points are awarded if the code is not wrapped in ```python```.

- **Correctness Score:** Similar to [14, 27], we use unit tests to verify the correctness of the code. Notably, while these works use a binary reward, we use the fraction of unit tests passed as the reward.

- **Safe Code:** To prevent the generation of unsafe code, we assign a reward of 0 if any blocked modules are used. These include os, sys, shutil, subprocess, socket, psutil, ctypes, pathlib, builtins, and __import__.

---

[8]`https://huggingface.co/datasets/Jiayi-Pan/Countdown-Tasks-3to4`
[9]`https://huggingface.co/datasets/ankner/math-500`
[10]`https://huggingface.co/datasets/KodCode/KodCode-Light-RL-10K`

## D.2 SFT Details

---

**Algorithm 2 Supervised Finetuning of LLaDA [30]**

---

**Require:** underlying unmasking predictor $f_\theta$, data distribution $p_{\text{data}}$, learning rate $\eta$

1: **repeat**
2:     Sample $(p_0, r_0) \sim p_{\text{data}}, \quad t \sim \mathcal{U}(0, 1)$         $\triangleright$ $p_0$ is the prompt and $r_0$ is the response
3:     Construct a partially masked response $r_t \sim q_{t|0}(r_t|r_0)$     $\triangleright$ $q_{t|0}$ is defined in Eq. (5)
4:     Calculate $\mathcal{L}(\theta) = -\frac{1}{t|r_0|} \sum_{i=1}^{|r_0|} \mathbb{1}[r_t^i = \texttt{mask}] \log f_\theta(r_0^i|p_0 \oplus r_t)$     $\triangleright$ $\oplus$ is concatenation
5:     $\theta \leftarrow \theta - \eta \nabla_\theta \mathcal{L}$
6: **until** Converged
7: **Return** $\theta$

---

Similarly, the SFT model also employs LoRA, with a rank of $r = 128$ and scaling factor $\alpha = 256$. We train with a sequence length of 4096 on 2 A6000 GPUs, using gradient accumulation over 4 steps and a per-device batch size of 1, yielding an effective batch size of 8. The optimizer and learning rate schedule match those used in *diffu*-GRPO, with a learning rate of 1e-5 and gradient clipping at 1.0. The SFT model was trained on the s1k dataset for 2460 steps, leaving 1% of the data for evaluation. A linear learning rate decay schedule was used, with no warmup. Our codebase contains further configuration details: `https://github.com/dllm-reasoning/d1`.

**Truncated Sequences** LLaDA-instruct is trained to generate full sentences, i.e., given any sequence length, it will always try to generate a complete sentence. However, due to the long sequence length of s1k, we had to truncate the dataset to have a maximum sequence length of 4096.

**Loss on PAD tokens** As discussed in Nie et al. [30], LLaDA needs to take a loss on the PAD tokens to be able to effectively terminate its generation. Additionally, to speed up training, we can pad the sequences in a batch to the longest sequence length in the batch. However, in GPU-constrained environments which use a small batch size, we find that padding to the longest datapoint in the batch is suboptimal since not enough PAD tokens are considered in the loss. Therefore, we choose to pad to the max length of the model.

**Dataset Difficulty** We find that there are some dynamics between the difficulty of the dataset and the model strength. In general, a weaker model needs a combination of easier and harder datapoints to scale reasoning. If we use an overly hard dataset, the performance degrades since the model easily overfits.

# E Qualitative Examples on GSM8k

We show qualitative results from all the models. Correct answers are in green, incorrect answers in red and aha-moments in blue. Notably, sometimes the model can reach the right answer via the wrong reasoning - such artifacts should reduce with a stronger base model.

## E.1 LLaDA-Instruct

When controlling the generation sequence length of dLLMs, it influences the reasoning style. For example, we see that with seq len = 512, compared to 128, the model produces more detailed step-by-step solutions with explicit mathematical formulas and enumerated calculations.

**Model:** LLaDA-8B-Instruct
**Generation Sequence Length:** 128
**Question:** A 76-star flag has three rows of 8 stars two rows of 6 stars and the rest are 5-star rows. How many rows of 5 stars are there altogether on the flag?

<reasoning> The flag has three stars in 8 rows, two stars in 6 rows, and the rest are 5-star rows. The total number of stars is the sum of the stars in the three8-star rows and the 6-star rows, which is 24 + 12 = 36 stars. Therefore, the remaining 76 - 36 = 40 stars are be placed in 5-star rows. Hence, the number of 5-star rows is 40/5. < /reasoning> <answer> 8 < /answer>

---

**Model:** LLaDA-8B-Instruct
**Generation Sequence Length:** 512
**Question:** A 76-star flag has three rows of 8 stars two rows of 6 stars and the rest are 5-star rows. How many rows of 5 stars are there altogether on the flag?

<reasoning> To determine the number of rows of 5 stars on the 76-star flag, we need to follow these steps:
1. Calculate the total number of stars in the three rows of 8 stars. 2. Calculate the total number of stars in the two rows of 6 stars. 3. Subtract the total number of stars in the first two rows from the total number of stars on the flag to find the number of stars in the 5-star rows. 4. Divide the number of stars in the 5-star rows by 5 to find the number of 5-star rows rows.
Let's go through each step in detail:
1. The total number of stars in the three rows of 8 stars is:

$$3 \times 8 = 24$$

2. The total number of stars in the two rows of 6 stars is:

$$2 \times 6 = 12$$

3. The total number of stars in the first two rows is:

$$24 + 12 = 36$$

4. The total number of stars on the flag is 76. Therefore, the number of stars in the 5-star rows is:

$$76 - 36 = 40$$

5. The number of 5-star rows is the number of stars in the 5-star rows divided by 5:

$$\frac{40}{5} = 8$$

So, the number of rows of 5 stars on the flag is 8 < /reasoning>

---

**Model:** LLaDA-8B-Instruct
**Generation Sequence Length:** 512
**Question:** A bakery produces 60 loaves of bread each day. Two-thirds of the loaves are sold in the morning and half of what is left is sold equally in the afternoon and evening. How many loaves of bread are sold in the afternoon?

<reasoning> To determine the number of loaves of bread sold in the afternoon, we need to follow these steps:
1. Calculate the number of loaves sold in the morning. 2. Determine the number of loaves remaining after the morning sale. 3. Calculate the number of loaves sold in the afternoon and evening. 4. Identify the number of loaves sold in the afternoon.

First, we calculate the number of loaaves sold in the morning. Since two-thirds of the loaves are sold in the morning, we have:
$$\frac{2}{3} \times 60 = 40$$
So, 40 loaves are sold in the morning.

Next, we determine the number of loaves remaining after the morning sale. The total number of loaves produced is 60, and 40 loaves are sold morning the morning, so the number of loaves remaining is:
$$60 - 40 = 20$$
So, 20 loaves are after the morning sale.

Then, we calculate the number of loaves sold in the afternoon and evening. Half of the remaining loaves are sold equally in the afternoon and evening, so the number of aves sold in the afternoon and evening is:
$$\frac{1}{2} \times 20 = 10$$
So, 10 loaves are sold in the afternoon and evening.

Finally, we identify the number of loaves sold in the afternoon. Since 10 loaves are sold in the afternoon and evening, and they are sold equally, the number of loaves sold in the afternoon is:
$$\frac{10}{2} = 5$$

So, 5 loaves are sold in the afternoon. $</$reasoning$><$answer$>$ $\boxed{5}$ $</$answer$>$

## E.2 SFT on Reasoning Traces from S1K

**Model:** LLaDA+SFT
**Generation Sequence Length:** 512
**Question:** A 76-star flag has three rows of 8 stars two rows of 6 stars and the rest are 5-star rows. How many rows of 5 stars are there altogether on the flag?

<reasoning> To determine the number of rows of 5 stars on the 76-star flag, we need to follow these steps:
1. Calculate the total number of stars in the three rows of 8 stars. 2. Calculate the total number of stars in the two rows of 6 stars. 3. Subtract the total number of stars in the first two rows from the total number of stars on the flag to find the number of stars in the remaining rows. 4. Divide the number of stars in the remaining rows by 5 to find the number of 5-star rows.

Let's go through each step in detail:
1. Calculate the number of stars in the three rows of 8 stars:

$$3 \times 8 = 24$$

So, there are 24 stars in the three rows of 8 stars.
2. Calculate the number of stars in the two rows of 6 stars:

$$2 \times 6 = 12$$

So, there are 12 stars in the two rows of 6 stars.
3. Subtract the total number of stars in the first two rows from the total number of stars on the flag:

$$76 - 24 - 12 = 40$$

So, there are 40 stars in the remaining rows.
4. Divide the number of stars in the remaining rows by 5 to find the number of 5-star rows:

$$\frac{40}{5} = 8$$

So, there are 8 rows of 5 stars.
However, we need to check if the number of rows is 8, as if there are 8 rows of 5 stars, the total number of stars would be:

$$8 \times 5 = 40$$

This matches the number of stars in the remaining rows, which is 40. Therefore, the number of rows of 5 stars is indeed 8.
Thus, the number of rows of 5 stars is:

$$\boxed{8}$$

< /reasoning><answer> 8 < /answer>

**Model:** LLaDA+SFT
**Generation Sequence Length:** 512
**Question:** A bakery produces 60 loaves of bread each day. Two-thirds of the loaves are sold in the morning and half of what is left is sold equally in the afternoon and evening. How many loaves of bread are sold in the afternoon?

<reasoning>To determine the number of loaves of bread sold in the afternoon, we need to follow these steps:
1. Calculate the number of loaves sold in the morning. 2. Determine the number of loaves remaining after the morning sales. 3. Calculate the number of loaves sold in the afternoon and evening. 4. Identify the number of loaves sold in the afternoon.
First. Calculate the number of loaves sold in the morning:

$$\text{Loaves sold in the morning} = \frac{2}{3} \times 60 = 40$$

Next. Determine the number of loaves remaining after the morning sales:

$$\text{Loaves remaining after morning sales} = 60 - 40 = 20$$

Then. Calculate the number of loaves sold in the afternoon and evening:

$$\text{Loaves sold in the afternoon and evening} = \frac{1}{2} \times 20 = 10$$

Since, half of the remaining loaves are sold equally in the afternoon and evening, we need to find the number of loaves sold in the afternoon..
Finally the number of loaves sold in the afternoon is:

$$\text{Loaves sold in the afternoon} = \frac{10}{2} = 5$$

However, this seems incorrect because the problem states that half of the remaining left is sold equally in the afternoon and evening, which means the should be:

$$\text{Loaves sold in the afternoon} = \frac{1}{2} \times 20 = 10$$

But since the problem states that half of the remaining left is sold equally in the afternoon and evening, the correct calculation should be:

$$\text{Loaves sold in the afternoon} = \frac{1}{2} \times 20 = 10$$

However, since the problem states that half of the remaining left is sold equally in the afternoon and evening, the correct calculation should be:

$$\text{Loaves sold in the afternoon} = \frac{1}{2} \times 20 = 10$$

< /reasoning> <answer> 10 < /answer>

### E.3  *diffu*-GRPO

**Model:** LLaDA+*diffu*-GRPO
**Generation Sequence Length:** 512
**Question:** A 76-star flag has three rows of 8 stars two rows of 6 stars and the rest are 5-star rows. How many rows of 5 stars are there altogether on the flag?

<reasoning> To determine the number of rows of 5 stars on the flag, we first need to calculate the total number of stars in the rows with 8 stars and 6 stars.
1. There are number of rows with 8 stars: $3 \times 8 = 24$ stars.
2. There are number of rows with 6 stars: $2 \times 6 = 12$ stars.
3. The these rows rows together have $24 + 12 = 36$ stars.
Next, we need the total number of stars on the flag, which is 76. We subtract the number of stars in the rows with 8 stars and 6 stars from the total number of stars:

$$76 - 36 = 40$$

So, there are 40 stars in the rows with 5 stars. To find the number of rows of 5 stars, we divide the number of stars in these rows by the number of stars per row:

$$\frac{40}{5} = 8$$

Therefore, there are 8 rows of 5 stars on the flag.
< /reasoning><answer> $\boxed{8}$ < /answer>

---

**Model:** LLaDA+*diffu*-GRPO
**Generation Sequence Length:** 512
**Question:** A bakery produces 60 loaves of bread each day. Two-thirds of the loaves are sold in the morning and half of what is left is sold equally in the afternoon and evening. How many loaves of bread are sold in the afternoon?

The bakery produces 60 loaves of bread each day. Two-thirds of the loaves are sold in the morning, so we calculate $\frac{2}{3} \times 60 = 40$ loaves sold in the morning. After the morning sales, there are $60 - 40 = 20$ loaves left. Half of these remaining loaves are sold equally in the afternoon and evening, so we calculate $\frac{1}{2} \times 20 = 10$ loaves sold in the afternoon.

Therefore< /reasoning><answer> $\boxed{10}$ < /answer>

## E.4  *d1*-LLaDA

**Model:** *d1*-LLaDA
**Generation Sequence Length:** 512
**Question:** A 76-star flag has three rows of 8 stars two rows of 6 stars and the rest are 5-star rows. How many rows of 5 stars are there altogether on the flag?

<reasoning> To determine the number of rows of 5 stars on a 76-star flag, we need to follow these steps:
1. Calculate the total number of stars in the three rows of 8 stars. 2. Calculate the total number of stars in the two rows of 6 stars. 3. Subtract the total number of stars in the first two rows from the total number of stars on the flag to find the number of stars in the remaining rows. 4. Divide the number of stars in the remaining rows by the number of stars per row to find the number of rows of 5 stars.
Let's go through each step in detail:
1. The total number of stars in the three rows of 8 stars is:

$$3 \times 8 = 24$$

2. The total number of stars in the two rows of 6 stars is:

$$2 \times 6 = 12$$

3. The total number of stars in the first two rows is:

$$24 + 12 = 36$$

4. Subtract the total number of stars in the first two rows from the total number of stars on the flag:

$$76 - 36 = 40$$

5. Divide the number of stars in the remaining rows by the number of stars per row to find the number of rows of 5 stars:

$$\frac{40}{5} = 8$$

Therefore, there are 8 rows of five stars on the flag. < /reasoning><answer> 8 < /answer>

**Question:** Jennifer's dog has 8 puppies 3 of which have spots. Brandon's dog has 12 puppies 4 of which have spots. What percentage of all the puppies have spots?

<reasoning> To determine the percentage of all the puppies that have spots, we need to follow these steps:
1. Calculate the total number of puppies from both dogs.
2. Calculate the total number of puppies with spots from both dogs.
3. Determine the percentage of puppies with spots.
First, let's find the total number of puppies from both dogs:
- Jennifer's dog has 8 puppies.
- Brandon's dog has 12 puppies.
- Total number of puppies = 8 + 12 = 20.
Next, let's find the total number of puppies with spots from both dogs:
- Jennifer's dog has 3 puppies with spots.
- Brandon's dog has 4 puppies with spots.
- Total number of puppies with spots = 3 + 4 = 7.
Now, we need to determine the percentage of puppies with spots. The formula for percentage is:

$$\text{Percentage} = \left( \frac{\text{Number of puppies with spots}}{\text{Total number of puppies}} \right) \times 100$$

Substituting the values we have:

$$\text{Percentage} = \left( \frac{7}{20} \right) \times 100$$

To simplify the fraction, we find the greatest common divisor (GCD) of 7 and 20, which is 1.

$$\frac{7}{20} = 7 \div 20$$

However, it seems there was a mistake in the calculation. Let's recheck the steps.
1. Total number of puppies = 8 + 12 = 20.
2. Total number of puppies with spots = 3 + 4 = 7.
3. Percentage of puppies with spots = $\left( \frac{7}{20} \right) \times 100$.
Let's calculate the percentage correctly:

$$\text{Percentage} = \left( \frac{7}{20} \right) \times 100 = \frac{7 \times 100}{20} = \frac{700}{20} = 35$$

The calculation answer. < /reasoning><answer> $\boxed{35}$ < /answer>

**Model:** *d1*-LLaDA
**Generation Sequence Length:** 512
**Question:** A bakery produces 60 loaves of bread each day. Two-thirds of the loaves are sold in the morning and half of what is left is sold equally in the afternoon and evening. How many loaves of bread are sold in the afternoon?

<reasoning> To determine the number of loaves of bread sold in the afternoon, we need to follow these steps:
1. Calculate the number of loaves sold in the morning. 2. Determine the number of loaves remaining after the morning sale. 3. Calculate the number of loaves sold in the afternoon and evening. 4. Identify the number of loaves sold in the afternoon.
Let's start by calculating the number of loaves sold in the morning. The bakery sells two-thirds of the loaves in the morning.

$$\text{Loaves sold in the morning} = \frac{2}{3} \times 60 = 40$$

Since we can't sell a fraction of whole loaves, we assume the problem implies for the integer part of the calculation.
So, the loaves sold in the morning are 40.
Next, we determine the number of loaves remaining after the morning sale.

$$\text{Loaves remaining after morning sale} = 60 - 40 = 20$$

Now, we calculate the number of loaves sold in the afternoon and evening. Half of the remaining loaves are sold equally in the afternoon and evening.

$$\text{Loaves sold in the afternoon and evening} = \frac{1}{2} \times 20 = 10$$

Since the problem asks for the number of loaves sold in the afternoon, we need the integer part of the division.

$$\text{Loaves sold in the afternoon} = \frac{10}{2} = 5$$

Therefore, the number of loaves of bread sold in the afternoon is $\boxed{5}$. < /reasoning>
<answer> $\boxed{5}$ < /answer>

