# OpenReview forum: "d1: Scaling Reasoning in Diffusion Large Language Models via Reinforcement Learning"
_NeurIPS.cc/2025/Conference — NeurIPS 2025 spotlight_

### Official Review · Reviewer_WpgT · 2025-07-02

**Clarity:** 3
**Significance:** 3
**Originality:** 4
**Rating:** 5
**Confidence:** 4

**Summary:**

This work is a first attempt at do RL-like finetuning of diffusion LLMs (dLLMs) on reasoning / math tasks. First, a couple important design choices are used to SFT LLaDA on the s1K reasoning dataset. The main challenge is how to do RL when we can't easily get the conditional logprob of the generated sequence due to how diffusion works. Mainly the logprob requires multiple passes of $f_\theta$ (inefficient) and cannot be decomposed as in regular AR models.

d1 proposes to estimate logprobs by perturbing the prompt and randomly masking each token with prob $p_{mask}$, then extracting the logprobs of completion $q$ in one step. For multiple PPO-epochs (termed gradient update iterations), the prompt is remasked each time which serves as a regularization and should give good logprobs in expectation.

Experiments are done on LLaDA 8B-instruct, further SFT on s1K. They train a separate model for each of 6 RL task across math, reasoning, coding, (e.g. MATH, Sudoku, HumanEval...) and vary the decoding length between 128,256,512. They find that SFT-ed on S1k may reduce performance initially but then RL training improves above baseline and SFT + diffusion GRPO is the best method. For evaluating RL-trained methods, they take the best performing model across runs, and across checkpoints. At 512 tokens, they observe "aha moments". They also generally find their method doesn't seem to improve for coding, likely owing to bad coding performance by LLaDA and s1k being focused on reasoning not coding.

**Questions:**

**addressing weaknesses**

can you train full finetuning LLaDA without LoRA or quantization on 4xH100?

the number of epochs of training and further, PPO-epochs of updates in GRPO seems wildly high compared to AR LLMs, is this just standard for diffusion? did you ablate any fewer number?
- s1 paper finetunes Qwen 2.5 for 5 epochs on s1k, you finetune for 20 epochs, did you try to ablate for fewer?

can you show final eval results instead of max across checkpoints for RL?
- you are currently choosing the best checkpoint for each task from step 600 to step 7700 (GSM8k) and this seems unfair compared to SFT / baseline

do you have any other seeds to give some level of statistical significance?
- even just on a single task, this would be helpful

**very important clarifications**

what are the specifics of your training in RL terminology e.g. for GSM8k
- how many completions per prompt $G$ do you use?
- what is the number of updates per generations (PPO epochs) $\mu$ ?
- how many episodes do you train on? (total number of generations i.e. total number of prompts * $G$)
- it would be ideal to be somewhat comparable in episodes to AR LLMs

**clarification**

does remasking q mean the reference logits need to be recalculated at each gradient update unlike traditional AR algorithms which calculate ref logprobs just once?

**Ethical Concerns:**

["NO or VERY MINOR ethics concerns only"]

**Final Justification:**

I believe this paper demonstrates a reasonable method for RL with diffusion LMs. The evaluations and technical details (small scale) are improved from initial submission but still lacking for an oral, in my opinion, but I believe it deserves to be part of the main conference and will definitely contribute to future work in this direction.

**Quality:**

2

**Strengths And Weaknesses:**

**strengths**
This work is very timely and pushes forward the boundaries of a very promising field. Diffusion LMs (DLMs) are quickly becoming a very strong alternative to autoregressive LLMs due to their efficiency and potentially using fewer inference steps at test time. Recent works like LLaDA have pushed the boundaries of pretraining but the actual use of these models as chatbots requires better postraining that previously available. To my knowledge, this is the first work tackling RL-style post-training for DLMs.

LLaDA is by no means a SOTA model compared to recent LLMs like Qwen3 so post-training them is quite difficult. This work manages to take a first step. Though this recipe seems quite lacking for full-scale post-training it is an important step towards SOTA DLM chatbots.

Currently, there are two major weaknesses. I believe this work should be accepted as long as they are clarified in text, even if not fully addressed with experiments. If both are addressed or the authors get strong results with a better evaluation protocol, this would be a clear accept. If the authors managed to make an effective post-trained chatbot using a single RL run across a mix of data, I would say this is a very strong paper and deserves an award.

**weaknesses**

The major weakness of the paper is the experimental setup has many questions of whether it will transfer to "real" situations. Specifically:

the work uses LLADA 8B quantized to 4 bits and trains only a LoRA
- most RLVR tasks for AR LLMs see much better performance with full finetuning and I think this would be a much more realistic training setup
- given that the authors have access to 8xH100 GPUs, is the training really so intensive that full finetuning is infeasible?

finetuning seems to use a smaller batch size and many more steps e.g. 7700 steps on GSM8k with total batch size ~100
- In [1](https://arxiv.org/pdf/2410.18252) they use Rho-1B and train Online DPO for 512 steps with batch size ~250 and in [2](https://arxiv.org/pdf/2410.01679) they train PPO with batch size 512 for 650 steps.
- This corresponds to nearly 2x the number of gradient steps, which could be reasonable if they use 2x updates per generation (i.e. $\mu = 2$)
- generally RL requires a larger batch size, learning rate, and fewer steps
- though this could

The second weakness is **evaluation protocol that is overly generous to their method**. They seem to choose the checkpoint with the highest eval score among all checkpoints for any given run and across 3 decoding lengths. This means for a task with inherent randomness, their evaluation chooses the max across many random decisions on the test set, which is a sort of optimization on the test set.

The evaluation method for their method should be fair in comparison to the baselines. I think that you can make things simpler by choosing the best generation length for each method and then comparing all methods after a specific number of steps. Otherwise, you should be using a validation set and after optimizing early stopping on the validation set, then you can show test set numbers.

Otherwise, HumanEval and MBPP don't make much sense as benchmarks to train on. They are far too small and only ever used for evaluation. The results with LightRL10k are a better metric and should really be used. I don't think its an issue that you don't have good coding results after s1k finetuning.

---

> ### Author Rebuttal · Authors · 2025-07-31
>
> We sincerely thank the reviewer for the thorough and constructive feedback. We appreciate your recognition that this is the first study to explore RL post-training for diffusion LLMs and that the topic is timely. Below we address each concern with new experiments and clarifications.
>
> > Q1: "The work uses LLaDA 8B quantized to 4 bits and trains only a LoRA. Most RL tasks for AR LLMs see much better performance with full finetuning and I think this would be a much more realistic training setup. Given that the authors have access to 8xH100 GPUs, is the training really so intensive that full finetuning is infeasible?"
>
> **R1:** Thank you for raising this point about training configuration. We acknowledge that full fine-tuning typically yields better performance than LoRA for AR LLMs, making this a crucial consideration for our work.
>
> The primary motivation for using LoRA with 4-bit quantization was resource constraints during method development. The majority of our experiments were conducted in academic settings with access only to A5000/A6000 GPUs, where full-parameter fine-tuning of 8B models was not feasible. Our 8xH100 cluster access was limited to the final experimental run and had restricted availability. We no longer have access to this cluster.
>
> We believe our results generalize to full fine-tuning settings, which represent a more powerful setup where we expect our method to show even greater benefits. To validate this hypothesis, we conducted new experiments comparing LoRA-based and full-parameter fine-tuning on GSM8K. Since image uploads are not permitted in rebuttals, we present the evaluation reward vs training steps results in the table below:
>
> **Table 1: Full fine-tuning vs LoRA fine-tuning of diffu-GRPO on LLaDA**
> | Training Step | LoRA 4-bit | Full FT 16-bit |
> |------|------------|----------------|
> | 250  | 77.2       | 78.2          |
> | 500  | 78.8       | 78.9          |
> | 750  | 79.2       | 79.9          |
> | 1000 | 79.8       | **80.3**      |
>
> For this experiment, we used a smaller policy gradient update (μ) value and larger batch size for stability in the full fine-tuning setting, while using same hyperparameters for the LoRA comparison. The results demonstrate that full fine-tuning achieves both higher peak accuracy and faster convergence compared to LoRA, confirming that our method scales effectively with stronger training configurations.
>
> > Q2: "The second weakness is evaluation protocol that is overly generous to their method. They seem to choose the checkpoint with the highest eval score among all checkpoints for any given run and across 3 decoding lengths. This means for a task with inherent randomness, their evaluation chooses the max across many random decisions on the test set, which is a sort of optimization on the test set." "Can you show final eval results instead of max across checkpoints for RL?"
>
> **R2:** We agree that our previous protocol may overestimate true generalization. In our revised experiments, we now report results from a fixed checkpoint (e.g., last checkpoint) in Table 2 below, and the results still show our conclusions hold.
>
> We note that our initial evaluation approach selected the best checkpoint across runs, but we applied this same protocol consistently to all RL runs, given the inherent variance at any fixed evaluation point.
>
> **Table 2: Final checkpoint evaluation results (same checkpoint number used for both RL runs)**
> | Dataset   | Gen Length | LLaDA | + DiffuGRPO | + SFT + DiffuGRPO |
> |-----------|------------|-------|-------------|-------------------|
> | GSM8K     | 128        | 68.7  | 71.2        | **72.5**         |
> | GSM8K     | 256        | 76.7  | 78.7        | **79.8**         |
> | MATH      | 128        | 26.0  | **32.3**    | 31.2             |
> | MATH      | 256        | 32.4  | 34.8        | **36.2**         |
> | Sudoku    | 128        | 11.7  | 14.2        | **20.9**         |
> | Sudoku    | 256        | 6.7   | **12.4**    | 12.7             |
> | Countdown | 128        | 20.7  | 25.7        | **29.6**         |
> | Countdown | 256        | 19.5  | 26.4        | **32.7**         |
>
> To further address generalization concerns, we highlight two additional experiments from our paper:
>
> (1) Combined training experiments (Table 2 in the paper), where we report results from a single fixed checkpoint across all tasks. Training a unified model across multiple tasks maintains strong performance while providing a more realistic evaluation protocol, as we did not select the best checkpoint for each individual task. The results demonstrate that training a single model for all tasks still outperforms both the baseline and SFT-only approaches.
>
> (2) Coding task evaluation (Table 3 in the paper on MBPP and HumanEval), where we consistently report performance from a single checkpoint.
>
> > Q3: "Do you have any other seeds to give some level of statistical significance?"
>
> **R3:** Thank you for highlighting the importance of statistical robustness in our evaluation. Yes, we have conducted experiments with various configurations and multiple seeds for GSM8K. We provide results averaged over three random seeds below, where we consistently observe improved performance over the original LLaDA, showing the robustness of our method under various settings rather than cherry-picking checkpoints. Additionally, in the paper's Table 1, we want to highlight that the improvements observed across multiple sequence lengths (128, 256, 512) during evaluation, despite training only at length 256, demonstrate that our method learns general reasoning strategies.
>
> **GSM8K Results (256 seq len, evaluated checkpoint at 7700 steps):**
> - Run 1: 81.1%, Run 2: 80.0%, Run 3: 80.4%
> - **Mean:** 80.5% +- 0.6%
>
> > Q4: "HumanEval and MBPP don't make much sense as benchmarks to train on. They are far too small and only ever used for evaluation. The results with LightRL-10k are a better metric and should really be used. I don't think it's an issue that you don't have good coding results after s1k finetuning."
>
> **R4:** To clarify, we do **not** train on HumanEval and MBPP. We only use those as evaluation datasets. We perform diffu-GRPO on LightRL-10k for coding tasks. This has been mentioned in Line 229, but we will clarify it further in the camera-ready version. Note that a single checkpoint is evaluated for all decoding lengths and tasks.
>
> > Q5: "The s1 paper finetunes Qwen 2.5 for 5 epochs on s1k, you finetune for 20 epochs, did you try to ablate for fewer?"
>
> **R5:** Thank you for this insightful question. We train longer because we observe that diffusion LLMs take more time to converge in SFT than autoregressive models. Because in each forward pass during training, the loss is only taken on the tokens that are randomly masked, whereas for AR the loss is applied to every token through teacher-foricng. This has been observed in other studies as well [1].
>
> We reserved 1% of the s1k dataset for evaluation and chose 20 epochs because the evaluation loss converges around this point. With fewer epochs, the diffusion LLM had not yet converged on the evaluation set, and we noticed that earlier checkpoints' evaluation accuracy was still climbing. Therefore, we chose to train until convergence to ensure fair comparison.
>
> > Q6: "What are the specifics of your training in RL terminology, e.g., for GSM8K?"
>
> **R6:** Here are the detailed RL training specifications:
>
> - **How many completions per prompt G do you use?**
>   We used 6 completions per prompt.
>
> - **What is the number of updates per generation (PPO epochs) μ?**
>   We used μ = 12 in the experiments.
>
> - **How many episodes do you train on? (total number of generations = total number of prompts × G)**
>   We used: per-device batch size = 6, GPUs = 8, gradient accumulation = 2, each prompt generates 6 completions. So each step processes 6 * 8 * 2 / 6 = 16 prompts. For GSM8K, we trained for 7700 steps with μ = 12, which gives us approximately 7700 ÷ 12 × 16 ≈ 10, 266 prompts, corresponding to ~ 1.5 epochs on the GSM8K training set.
>
> > Q7: does remasking q mean the reference logits need to be recalculated at each gradient update unlike traditional AR algorithms which calculate ref logprobs just once?
>
> Yes, this is a detailed observation. In AR RL algorithms, reference log probabilities are calculated once during policy gradient iterations. However, with our remasking approach, the log probabilities are conditioned on the specific masking pattern, requiring us to compute μ sets of log probabilities for the reference policy to ensure fair calculations. We view this as a form of data augmentation compared to AR models.
>
> We hope these clarifications and additional experiments address your concerns. We are happy to answer any further questions you might have and appreciate your valuable feedback in improving our work.
>
> [1] Prabhudesai, M., Wu, M., Zadeh, A., Fragkiadaki, K., & Pathak, D. (2025). Diffusion Beats Autoregressive in Data-Constrained Settings. arXiv preprint arXiv:2507.15857.

---

> > ### Author Response · Authors · 2025-08-04
> >
> > Dear Reviewer WpgT,
> > Thanks again for your diligent feedback on our work! We've carefully addressed your concerns with new experiments and clarifications, including full fine-tuning vs lora comparisons, revised evaluation protocols using fixed checkpoints, multi seeds statistical validation, detailed RL training specifications, and experimental setup clarifications. We believe these additions strengthen our work and address the evaluation protocol concerns you raised.
> >
> > Could you please let us know if these clarifications and new results address your concerns? We would be happy to discuss any remaining questions. Thank you!

---

> > > ### Author Response · Authors · 2025-08-06
> > > **gentle reminder**
> > >
> > > Dear Reviewer WpgT,
> > >
> > > We thank you again for your diligent review and feedback. We have tried our best to address your comments through the rebuttal period with new experiments and clarifications. We believe these address the evaluation protocol concerns you raised. Since the author discussion period will end soon, please let us know if there are any more questions that can be addressed. Thank you!

---

> > > > ### Comment · Reviewer_WpgT · 2025-08-08
> > > > **Thank you, Updating Score**
> > > >
> > > > thank you for addressing my issues and comments, I will increase my score and recommend acceptance
> > > >
> > > > **full finetuning LLaDA**
> > > > thank you for these results, they are encouraging
> > > >
> > > > **final ckpt evaluation methodology**
> > > > thank you for these results, since they are equally convincing, please change the main paper results to these as this is a more standard eval methodology
> > > >
> > > > **avg over 3 seeds** thank you for the results, they are encouraging. could be worth adding to the appendix, but not necessary
> > > >
> > > > **clarifications and details** these are all very useful and would make great additions to the appendix

---

### Official Review · Reviewer_rbC8 · 2025-07-03

**Clarity:** 1
**Significance:** 2
**Originality:** 3
**Rating:** 4
**Confidence:** 4

**Summary:**

The paper introduces an efficient, critic-free policy-gradient method for fine-tuning masked diffusion models. The method's efficiency stems from two main aspects: (1) it does not require a parameterized critic or value function; instead, it employs group statistics (similar to GRPO) for value estimation, and (2) the policy updates are conducted using one-shot log-probability estimation rather than requiring full trajectories. Additionally, the paper introduces random masking of the prompt as a regularization technique. To demonstrate the efficacy of the method, experiments are performed on question answering style reasoning tasks (GSM8k and Math500) and iterative reasoning tasks like Sudoku and Countdown.

**Questions:**

1. The impact of using random masking in the prompt is still unclear, especially for moderate values of $\mu$ like 5 to 10. To substantiate the claim in lines 52-55, more direct evidence is necessary. Would it be possible to measure the overall efficiency gain using the number of total forward and backward passes performed during the entire training? For example, if the budget is fixed in terms of total forward/backward passes, how does the accuracy change when one varies the probability of prompt masking?


2. What is the exact format in which countdown and sudoku puzzles are provided to the model? Could you give one example input-output pair for each?

    - As a follow-up, why is the accuracy for sudoku falling as the generation length increases?

3. **Experimental Details**:

    1. Regarding the “+SFT” row in Table 1 for GSM8k and Math500:

        - How many examples did the model see? According to line 601, it should be 2460\*16 = 39360\. For a dataset of 1k examples, this is roughly 40 epochs. But line 194 says the SFT model was trained for 20 epochs. Am I missing something?

    2. Similar question for “+SFT+Diffu-GRPO” and “+Diffu-GRPO”. Line 541 states that the RL training was performed for 7700 and 6600 steps, respectively, with an effective batch size of 8 x 6 = 48. Without further details, I assume it was the same for “+SFT+Diffu-GRPO” as well as “+Diffu-GRPO”. The difference in the number of training steps for the "+SFT" and "+Diffu-GRPO" makes them difficult to compare. Could you provide the reward curves for SFT checkpoints with respect to the number of gradient updates, similar to Figure 7?

**Ethical Concerns:**

["NO or VERY MINOR ethics concerns only"]

**Final Justification:**

There was a lengthy discussion, and the authors clarified many minute details that were confusing. The paper still needs careful editing so that these questions do not arise.

**Limitations:**

yes

**Quality:**

2

**Strengths And Weaknesses:**

**Clarity**: The method is explained clearly, but the experimental details are spread out across the paper, including the appendix, and some details seem to conflict. As it stands, it’s hard to grasp the significance of the results (see questions titled “Experimental Details”) with these details missing.

**Quality**: The paper makes interesting points, but some claims could benefit from clearer support:

1. The results in the paper may not directly serve as a good recipe for finetuning a masked diffusion model. For instance, it’s unclear if the “+SFT” and “+diffu-GRPO' results in Table 1 are directly comparable (see the questions section for details).
2. Figure 5 suggests that random masking of the prompt allows higher $\mu $, but its overall impact on performance within a given budget isn’t fully clear (refer to question 1 below).
3. Comparisons with autoregressive models could be clarified. To the best of my understanding, based on available training reports, Llama 3 and Qwen 2.5 were not explicitly trained on GSM8k, whereas d1-LlaDa was fine-tuned on GSM8k to achieve 82.1% accuracy reported in Figure 3. For a more direct comparison, fine-tuning AR models on GSM8k using SFT or RL would be ideal.

**Significance:** Although using RL to fine-tune masked diffusion models is an interesting research direction, concerns about the quality and clarity of the submitted draft diminish its significance.

**Originality**: The paper proposes a GRPO-like training procedure for fine-tuning masked diffusion models. It also proposes the use of random masking of the prompt as a regularization technique. Both contributions are original.

---

> ### Author Rebuttal · Authors · 2025-07-31
>
> We thank Reviewer rbC8 for their detailed feedback. We appreciate your recognition of the originality of our method and constructive suggestions for improving clarity and evaluation. Below we address each concern with clarifications and additional experimental details.
>
> > Q1: Regarding the "+SFT" row in Table 1 for GSM8k and Math500:
> How many examples did the model see? According to line 601, it should be 2460*16 = 39360. For a dataset of 1k examples, this is roughly 40 epochs. But line 194 says the SFT model was trained for 20 epochs.
>
> Thank you for catching this typo. We used an effective batch size of 8 (not 16) and reserved 1% of data for evaluation (990 training examples). With HuggingFace Trainer's integer division, this gives 123 steps per epoch (990÷8=123). Over 20 epochs, this totals 2,460 steps. We will clarify this calculation in the revision.
>
> > Q2.1: The different training steps for "+SFT" and "+Diffu-GRPO" make them difficult to compare. Could you provide reward curves for SFT checkpoints with respect to gradient updates?
>
> Thank you for raising this important question. We acknowledge that the varying training steps across methods require clarification and appreciate the opportunity to address this concern.
>
> **Different step counts reflect principled, convergence-based stopping criteria:** The different step counts in our experiments are not arbitrary choices but reflect task-specific convergence approaches. For RL, since different tasks have varying difficulty levels, they converge at different training steps. We monitored training reward curves and applied early stopping when the reward curve plateaued, leading to our reported 7,700 steps for GSM8K vs. 6,600 steps for MATH in RL training. For SFT, we reserved 1% of training data for evaluation and monitored evaluation loss to determine the convergence point when the evaluation loss flattened.
>
> To clarify any confusion, the "+SFT+Diffu-GRPO" and "+Diffu-GRPO" experiments are trained for the same number of RL steps in our experiments.
>
> > Q2.2: Could you provide the reward curves for SFT checkpoints with respect to the number of gradient updates, similar to Figure 7?
>
> Yes, we provide the SFT learning curve for MATH500 in the table below (as we are not allowed to upload images in rebuttal):
>
> | Step | 300 | 600 | 1200 | 1800 | 2300 | 2460 |
> |------|-----|-----|------|------|------|------|
> | MATH500 Accuracy | 29.8 | 29.6 | 31.0 | 32.3 | 32.4 | 32.6 |
>
> - We note that **it is not necessary for SFT and RL to use the exact same training steps for comparison**, as SFT and RL employ fundamentally different learning paradigms. SFT uses token-level cross-entropy loss with sparse learning signals (only on randomly masked tokens), while RL uses sequence-level reward optimization.
>
> - Additionally, the RL step count represents the number of policy gradient updates per iteration, which means the same data sample can be updated multiple times. In our case, we used μ=12, meaning we perform 12 gradient updates on the same prompt's completion groups. Furthermore, SFT might require significantly more epochs to converge because each forward pass only provides learning signals on masked tokens, whereas RL provides sequence-level feedback. This phenomenon has been observed in other studies comparing autoregressive and diffusion LLMs [1].
>
> - Therefore, we believe convergence-based stopping criteria provide a more principled approach than requiring identical training steps for SFT and RL.
>
> - Additionally, we do not aim to argue that RL is universally superior to SFT in our work. Rather, our focus is on developing effective RL post-training methods for diffusion LLMs. We found that SFT-initialized models provide better exploration foundations for subsequent RL training—a recipe that has proven effective in prior works such as DeepSeek-R1, which also employs SFT as a cold start followed by RL training. This motivates our exploration of this approach for diffusion LLMs as well. Our reporting of both "+Diffu-GRPO" and "+SFT+Diffu-GRPO" configurations demonstrates that these methods serve complementary purposes in our post-training pipeline, with the combined approach consistently yielding the best results.
>
> > Q3: To substantiate the claim in lines 52-55, more direct evidence is necessary. Would it be possible to measure the overall efficiency gain using the number of total forward and backward passes performed during the entire training?
>
> Thank you for this important question about substantiating our efficiency claims. We believe our ablation experiments in Figure 5 already address these concerns, but we'd like to provide additional clarification with new tables of results.
>
> - **Wall clock time corresponds to total forward/backward passes:** Figure 5's rightmost panel shows wall clock time, which directly corresponds to the total forward and backward passes during training. To make this relationship clearer, we have calculated the exact forward pass counts for different μ values. In our experimental setup, each completion has 128 forward steps for denoising, and after sampling, μ gradient updates are performed on them. The table below demonstrates that higher μ values achieve superior correctness rewards with significantly fewer forward passes:
>
> | Global_Step | μ=2_forward passes | μ=2_reward | μ=12_forward passes | μ=12_reward | μ=24_forward passes | μ=24_reward |
> |-------------|---------------------|-------------|----------------------|--------------|----------------------|--------------|
> | 1000        | 64.5k               | 0.5         | 11.2k                | 1.0          | 5.8k                 | 1.1          |
> | 2000        | 129k                | 0.8         | 22.3k                | 1.2          | 11.7k                | 1.4          |
> | 3000        | 193.5k              | 0.8         | 33.5k                | 1.2          | 17.5k                | 1.3          |
> | 4000        | 258k                | 0.9         | 44.7k                | 1.3          | 23.3k                | 1.3          |
> | 5000        | 322.5k              | 0.9         | 55.8k                | 1.3          | 29.2k                | 1.3          |
> | 6000        | 387k                | 1.0         | 67k                  | 1.4          | 35k                  | 1.2          |
> | 7000        | 451.5k              | 1.0         | 78.2k                | 1.4          | 40.8k                | 1.1          |
>
> - Additionally, the left three panels of Figure 5 show reward progression versus the number of completions generated during training, which reflects computational budget since online sampling dominates the RL training time. These results consistently demonstrate that random masking enables stable scaling to higher μ values (12) while maintaining stable learning dynamics.
>
> - Other work has also found that diffusion language models outperform autoregressive models through data augmentation via random masking [1]. This aligns with our hypothesis that random masking serves as a form of regularization, supporting our efficiency claims in lines 52-55.
>
> [1] Prabhudesai, Mihir, et al. "Diffusion Beats Autoregressive in Data-Constrained Settings." arXiv:2507.15857(2025).
>
> > Q4: If the budget is fixed in terms of total forward/backward passes, how does the accuracy change when one varies the probability of prompt masking?
>
> We note that the prompt masking probability does not affect the computation usage, but it affects the learning. To answer your question, we provide evidence in Figure 6 of our paper. Light masking (pmask = 0.1, 0.3) consistently improves training stability and final performance versus no masking (pmask = 0.0), while higher rates (0.5, 0.7) introduce instability. This supports our regularization claims.
>
> > Q5.1: What is the exact format in which countdown and sudoku puzzles are provided to the model? Could you give one example input-output pair for each?
>
> Please find below the prompt formats:
>
> **Sudoku format:**
> ```
> Please solve the following 4x4 Sudoku puzzle...
> Solve the following Sudoku puzzle:
> 0002403020033024
> ```
>
> **Countdown format:**
> ```
> Using only the provided numbers, create an arithmetic expression...
> Numbers: [49, 55, 53]
> Target: 51
> ```
>
> Please note that since we use the instruct model, there are additional special tokens and a chat template as well—however, we omit them from this example for clarity.
>
> > Q5.2: Why is the accuracy for sudoku falling as the generation length increases?
>
> The drop in accuracy with increased generation length is not specific to diffuGRPO—we observe the same trend with LLaDA itself as shown in Table 1. We hypothesize that this may be due to its limited long-horizon planning capabilities when the initialized response sequence is long. Additionally, it may stem from the model's general difficulty in planning on search-based tasks like Sudoku. Another possible factor is the decoding strategy, such as the semi-AR decoding and block size choices, which are beyond the scope of our current study.
>
> > Q6: Comparisons with autoregressive models could be clarified.
>
> - We appreciate the reviewer's point about the need for a more direct comparison with AR models. However, we would like to argue that it is not possible to obtain a 100% fair comparison since we do not know the training details of these AR models.
>
> - Our goal is not claim we surpass AR models outright, but to establish that diffusion LLMs are viable and competitive, even without the large-scale post-training that these SOTA AR models have undergone. We mainly want to demonstrate that our RL-trained diffusion models can achieve performance comparable to established AR benchmarks. As this will show the potential of applying our methods on a larger scale with stronger base models—akin to what has been done with Qwen2.5 and LLaMA3—can further push frontier dLLM performance.
>
> We hope these clarifications and additional experiments address your concerns. We are happy to answer any further questions you might have.

---

> > ### Author Response · Authors · 2025-08-04
> > **gentle reminder**
> >
> > Dear Reviewer rbC8,
> >
> > Thanks again for your diligent feedback on our work! We've carefully addressed your comments with new experiments and clarifications, including detailed explanations of SFT and RL convergence criteria, SFT reward curves across training steps, analysis of efficiency gains from random masking measured by forward/backward pass counts, format examples for sudoku and countdown tasks, and clarifications addressing your evaluation concerns.
> >
> > We hope these address your concerns. Could you please let us know if you have further questions? We would be happy to discuss further. Thank you!

---

> > > ### Comment · Reviewer_rbC8 · 2025-08-05
> > > **Thanks for providing detailed clarifications**
> > >
> > > Thanks for answering my questions. I have a follow-up question.
> > >
> > > 1. To better understand the impact of moving from SFT to RL, I'm examining Figure 7 and Table 1. According to Table1, the final accuracy difference between Base, +SFT vs +SFT+diffu-GRPO for GSM8k is 76 vs 78.8 vs 81.2 (roughly 3-5% increase), and with accuracy around 76, the initial reward should at least start around 1.52 (based on the breakdown of the reward score provided in Appendix D.1.1). However, in Figure 7 (1), the rewards start at almost zero and ends at around 1.57. This confuses me. What am I missing? This is one of the reasons I requested retro-computed **reward values** (not accuracy values) at various stages of the SFT training. I have a similar question for Math500.
> > >
> > > 2. Regarding comparisons with AR models: while I understand that you wish to demonstrate that the RL-trained Diffusion models can be competitive with AR models, which they appear to be, it is still important to clearly state the settings under which the comparison is made. Therefore, at the very least, a line in the caption or a footnote is warranted to clarify this in the subsequent revision of the paper.

---

> ### Author Response · Authors · 2025-08-05
> **thank you for the follow up question**
>
> Dear Reviewer rbC8,
>
> Thank you for the detailed follow-up question!
>
> 1. The step 0 reward in Figure 7 represents the SFT model's performance under our strict reward function parsing. The reason the GSM8K reward starts from zero is due to our reward function's strict parsing requirements: we use a correctness function that only extracts content inside `<answer></answer>` tags and uses everything inside the XML tag as the final answer to compare with ground truth answers, which is usually an integer. Below are examples from our SFT model that illustrate why most outputs receive zero rewards initially:
>
> Example 1:
> ```
> Generation: ...<answer>Therefore, the remainder when (n+2)(n+4)(n+6) is divided by 7 is \boxed{0}.</answer>
> Correctness reward = 0 (too many words inside the tag for exact matching)
> ```
>
> Example 2:
> ```
> Generation: ...Therefore, the smallest positive integer n is n = 4</answer>\boxed{4}
> Correctness reward = 0 (missing opening XML tag: '<answer>')
> ```
>
> As you can see, the SFT model struggles with following the XML prompt template correctly. Additionally, the reward curves represent batch-averaged values across multiple samples, so when most outputs initially fail format compliance, the averaged reward starts near zero even if some individual samples might achieve partial correctness.
>
> Below are the reward values of the exact parsing function for GSM8K on 500 GSM8K eval examples:
>
> Table 1: Correctness reward function of SFT checkpoints
>
> | Step          | 600   | 1200  | 1800  | 2460  |
> |---------------|-------|-------|-------|-------|
> | Avg Reward    | 0.020 | 0.016 | 0.020 | 0.016 |
>
> These reward values are not meaningful as they primarily reflect formatting failures rather than correctness due to the strict XML tag parsing requirements (`<answer></answer>` tags with exact matching).
>
> The benefit of this strict parsing approach is that once the model learns proper format compliance and achieves consistent format rewards, it can then focus on improving correctness. We believe using different parsers for RL training versus evaluation fosters faster and more stable learning.
>
> Our approach aligns with other works in the field. Unsloth AI's GRPO notebooks for GSM8K (for example, with Llama 3.1) follow almost the same reward setup, where they extract from XML tags and perform exact matching. Recent work (arXiv:2503.01307) similarly implements format rewards with reward curves beginning near zero.
>
> For fairness in our accuracy evaluations, we maintain a generous accuracy parser for all models that can extract answers from various formatting styles. This ensures that almost all responses from all models get properly parsed and evaluated consistently, which is why we can report meaningful accuracy percentages even when RL training rewards start low.
>
> 2. We fully agree with your point about clearly stating the comparison settings. We will add a clarification note in the caption of the comparison figure to make this clearer.
>
> Thank you for the suggestions, and we hope these clarifications address your concerns.

---

> > ### Author Response · Authors · 2025-08-06
> > **gentle reminder**
> >
> > Dear Reviewer rbC8,
> >
> > We thank you again for your diligent review and the follow-up feedback. We have addressed your comments with clarifications and results above. Since the discussion is about to end, please let us know if we can address any more concerns before the author discussion period is over. Thank you!

---

> > > ### Comment · Reviewer_rbC8 · 2025-08-07
> > > **No further questions**
> > >
> > > Thanks for answering all my questions. I have no further questions.

---

> > > > ### Author Response · Authors · 2025-08-07
> > > >
> > > > Hi reviewer rbC8,
> > > >
> > > > We’re glad our answers addressed your questions. Since you feel the questions have been resolved, would you consider updating your score to reflect that?
> > > >
> > > > We’ll be sure to incorporate your suggestions into the paper. Please let us know if there’s anything else we can provide. Thanks!

---

> > > > > ### Comment · Reviewer_rbC8 · 2025-08-08
> > > > >
> > > > > I have updated my score.

---

### Official Review · Reviewer_SVRy · 2025-07-03

**Clarity:** 2
**Significance:** 3
**Originality:** 3
**Rating:** 4
**Confidence:** 4

**Summary:**

This paper presents d1, a framework that adapts pretrained masked diffusion LLMs (dLLMs) for reasoning via masked supervised finetuning and a novel RL algorithm, diffu-GRPO. Empirical results show d1 improves reasoning on math and planning tasks.

**Questions:**

1. What sampling temperature is used during RL? Or is greedy decoding applied?
2. Why choose an RL generation length of 256 when the SFT context window is 4096? What motivates this design?

**Ethical Concerns:**

["NO or VERY MINOR ethics concerns only"]

**Limitations:**

Yes

**Quality:**

3

**Strengths And Weaknesses:**

**Strengths**

1. Introducing RL for diffusion LLMs is a necessary and interesting direction.
2. d1 significantly boosts reasoning performance over prior dLLMs.
3. The improvements generalize across different output lengths.

**Weaknesses**

1. Details of methods that are listed in the appendix are missing in main-text, which may cause misunderstanding. For example, semi-autoregressive decoding is used, and how s1k data is chunked given its average length > 4096.
2. Line 228 asserts diffu-GRPO scales to multi-task settings “without sacrificing accuracy” vs. per-task results, but MATH length 128 drops from 33.8 to 29.8, which is quite significant, contradicting that claim.
3. In d1's design, randomly masking the prompt to estimate log-probs may introduce extra noise, since there is no clean conditioning signal.

---

> ### Author Rebuttal · Authors · 2025-07-31
>
> We sincerely thank the reviewer for their thoughtful and constructive feedback on our paper. We appreciate your recognition of the novelty and significance of introducing RL for diffusion LLMs, as well as your acknowledgment of d1's substantial performance improvements. Below we address your questions and concerns:
>
> > Q1: Details of methods that are listed in the appendix are missing in main-text, which may cause misunderstanding. For example, semi-autoregressive decoding is used, and how s1k data is chunked given its average length > 4096.
>
> We agree that key methodological details should be moved from the appendix to the main paper. The details were relegated to the appendix solely due to space constraints. We will move these crucial implementation details into the main body to ensure clarity and prevent potential misunderstandings.
>
> > Q2: Line 228 asserts diffu-GRPO scales to multi-task settings "without sacrificing accuracy" vs. per-task results, but MATH length 128 drops from 33.8 to 29.8, which is quite significant, contradicting that claim.
>
> - Thank you for raising this question. We acknowledge that our phrasing in line 228 was imprecise. When we stated that diffu-GRPO scales to multi-task settings "without sacrificing accuracy," we intended to convey that the framework maintains competitive performance when training across multiple tasks simultaneously, rather than claiming identical performance to single-task training. The drop represents a trade-off when mixing diverse data distributions in multi-task learning. However this multi-task settings still demonstrates that our approach achieves reasonable multi-task generalization without catastrophic performance degradation.
>
> - We will revise this claim to more accurately reflect the multi-task learning trade-offs.
>
> > Q3: In d1's design, randomly masking the prompt to estimate log-probs may introduce extra noise, since there is no clean conditioning signal.
>
> This is an insightful observation about our design choice. While random prompt masking could theoretically introduce noise by eliminating clean conditioning signals, our empirical results demonstrate higher learning efficiency and stable performance with this approach. As shown in our ablation study in Figure 5, random masking serves as a form of data augmentation that prevents dLLMs from overfitting too quickly in policy gradient iterations. This finding is also echoed by other works [1], which demonstrate that random masking of dLLMs brings data augmentation effects.
>
> We hypothesize that making the conditioning signal slightly noisier makes the task harder, leading to improved learning. And we can trade off the accuracy of the log-probability for more efficient learning. Though this is an interesting point to study further, an intriguing follow-up could explore whether there is an optimal balance in this trade-off by introducing schedules such that the model can use masking to support higher learning rates for fast initial learning, then stabilize and use no prompt masking with smaller learning rates for final annealing.
>
> [1] Prabhudesai, Mihir, et al. "Diffusion Beats Autoregressive in Data-Constrained Settings." arXiv preprint arXiv:2507.15857 (2025).
>
> > Q4: What sampling temperature is used during RL? Or is greedy decoding applied? Why choose an RL generation length of 256 when the SFT context window is 4096? What motivates this design?
>
> We employed a sampling temperature of 0.7 during RL training to encourage diversity within GRPO sampling groups. The choice of sequence length 256 for RL training, despite the 4096 SFT context window, was primarily driven by computational efficiency considerations. Online RL training is bottlenecked by generation time, and the 256-token length provided a reasonable training duration (~1 day to convergence in our compute setup) while still demonstrating strong generalization to longer sequences during inference.
>
> We hope these clarifications address your concerns while highlighting the potential of our approach.

---

### Official Review · Reviewer_3Dmw · 2025-07-03

**Clarity:** 4
**Significance:** 4
**Originality:** 4
**Rating:** 5
**Confidence:** 4

**Summary:**

The paper introduces a method of applying GRPO to post training stage  of diffusion LLMs. It was unclear that whether the gains in autoregressive modelling through techniques like GRPO could be applied to diffusion modelling or not. This paper provides a first step in this direction by applying GRPO using a one step log-probability estimation technique. The results show comparable performance to auto-regressive LLMs.

**Questions:**

what are the losses incurred or side effects of approximating log-probs? One way to analyse this could be using the same method on autoregressive transformer model v/s using the exact log-probs. Nevertheless, I think that it is a good work!

**Ethical Concerns:**

["NO or VERY MINOR ethics concerns only"]

**Final Justification:**

I believe that the paper is good and should be accepted to the conference. The authors answered my questions and my final rating is accept

**Limitations:**

yes

**Quality:**

4

**Strengths And Weaknesses:**

Strengths:
- This is the first application of GRPO to diffusion models for post training with efficient one step log probability estimation
- d1 finetuned models perform comparable to transformer models of similar sizes.

Weaknesses:
- The paper acknowledges that exact log-prob calculation is difficult for dLLMs, hence it uses a one-step estimation based on mean field approximation. The paper does not discuss the side-effects of this approximation.

---

> ### Author Rebuttal · Authors · 2025-07-31
>
> We would like to thank you for the thoughtful and encouraging review. We’re glad the quality and clarity of this paper appealed to you!
>
> > Q1: The paper acknowledges that exact log-prob calculation is difficult for dLLMs, hence it uses a one-step estimation based on mean field approximation. The paper does not discuss the side-effects of this approximation.
>
> The problem with using a single-step log-prob estimation is that the values can be noisy due to higher variance. However, we noticed that the per-token log-probs can be stabilized with as few as 4-5 Monte Carlo samples using LLaDA's estimation algorithm. This might be due to the masked formulation and the training of the CE loss. While that is still intractable to compute, empirically, we find that the variance with a single sample did not significantly affect performance on the tasks we experimented with, as the training stabilizes even with high values of policy gradient iterations (μ). We believe our method can also be modified to reduce variance, such as by reducing the masking rate applied to the prompt and enlarging the batch size.
>
> > Q2: what are the losses incurred or side effects of approximating log-probs? One way to analyse this could be using the same method on autoregressive transformer model v/s using the exact log-probs. Nevertheless, I think that it is a good work!
>
> Thank you! We would like to clarify that the same method cannot be used on an autoregressive model since there is no notion of estimating the log-probs - the value we get will be the exact log-prob due to its training objective for autoregressive transformers.

---

> > ### Comment · Reviewer_3Dmw · 2025-08-04
> >
> > Thanks for answering my questions. My rating remains the same.

---

### Decision · Program_Chairs · 2025-09-17

**Decision:**

Accept (spotlight)

**Comment:**

I recommend accepting this paper. The authors present d1, the first framework applying reinforcement learning to diffusion language models (dLLMs) for improved reasoning. Their key contributions include a masked supervised fine-tuning technique and diffu-GRPO, a novel critic-free policy gradient algorithm that efficiently estimates log-probabilities in one step. All four reviewers endorsed acceptance, noting the work's originality and significance. The authors effectively addressed reviewer concerns with additional experiments showing full fine-tuning results, fixed checkpoint evaluations, and multi-seed validation. Results demonstrate significant improvements over baseline dLLMs on mathematical reasoning and planning tasks, approaching the performance of comparable autoregressive models. This work represents an important advancement in extending RL techniques beyond traditional autoregressive generation paradigms.